# Climate change vs. socio-economic development: Understanding the future South Asian water gap

René R. Wijngaard[1,4], Hester Biemans[2], Arthur F. Lutz[1], Arun B. Shrestha[3], Philippus Wester[3], and Walter W. Immerzeel[1,4]

[1]FutureWater, Costerweg 1V, 6702 AA, Wageningen, The Netherlands
[2]Water and Food research group, Wageningen University & Research, PO Box 47, 6700 AA Wageningen, The Netherlands
[3]International Centre for Integrated Mountain Development, GPO Box 3226, Khumaltar, Kathmandu, Nepal
[4]Utrecht University, Department of Physical Geography, PO Box 80115, 3508 TC, Utrecht, The Netherlands

*Correspondence to*: René R. Wijngaard (r.wijngaard@futurewater.nl / r.r.wijngaard.uu@gmail.com)

**Abstract.** The Indus, Ganges, and Brahmaputra (IGB) river basins provide about 900 million people with water resources used for agricultural, domestic, and industrial purposes. These river basins are marked as "climate change hotspot", where climate change is expected to affect monsoon dynamics and the amount of meltwater from snow and ice, and thus the amount of water available. Simultaneously, rapid and continuous population growth, and strong economic development will

likely result in a rapid increase in water demand. Since quantification of these future trends is missing, it is rather uncertain how the future South Asian water gap will develop. To this end, we assess the combined impacts of climate change and socio-economic development on the future "blue" water gap in the IGB until the end of the 21st century. We apply a coupled modelling approach consisting of the distributed cryospheric-hydrological model SPHY, which simulates current and future upstream water supply, and the hydrology and crop production model LPJmL, which simulates current and future

downstream water supply and demand. We force the coupled models with an ensemble of eight representative downscaled General Circulation Models (GCMs) that are selected from the RCP4.5 and RCP8.5 scenarios, and a set of land use and socio-economic scenarios that are consistent with the Shared Socio-economic Pathway (SSP) marker scenarios 1 and 3. The simulation outputs are used to analyse changes in the water availability, supply, demand, and gap. The outcomes show an increase in surface water availability towards the end of the 21st century, which can mainly be attributed to increases in

monsoon precipitation. However, despite the increase in surface water availability, the strong socio-economic development and associated increase in water demand will likely lead to an increase in the water gap during the 21st century. This indicates that socio-economic development is the key driver in the evolution of the future South Asian water gap. The transgression of future environmental flows will likely be limited with sustained environmental flow requirements during the monsoon season and unmet environmental flow requirements during the low flow season in the Indus and Ganges river

basins.

# 1. Introduction

Freshwater resources are essential for hundreds of millions of people living in South Asian river basins. The Indus, Ganges, and Brahmaputra (IGB) river systems provide about 900 million people and the world's largest irrigation scheme (i.e. Indus Basin Irrigation System's (IBIS)) with water, which is mainly used for agricultural (e.g. irrigation), domestic (e.g. drinking water supply), and industrial purposes, (FAO, 2012; Klein Goldewijk et al., 2010; Rasul, 2014; Shrestha et al., 2013).

The water supply in the IGB is mainly dominated by two different components: locally pumped groundwater and surface water supplied by irrigation canals. Groundwater is an important water supplier for the agricultural sector with contributions of about 64% and 33% to the total irrigation water supply in India and Pakistan, respectively (Biemans et al., 2016; Siebert et al., 2010). Surface water is supplied by irrigation canals that are diverted from rivers and reservoirs, and constitute of direct rainfall runoff, meltwater from upstream located ice and snow reserves, and baseflow. Meltwater is the largest constituent of the total annual surface flow in the western part of the IGB, where the amount of winter precipitation is substantial and the largest ice reserves are present (Bookhagen and Burbank, 2010; Immerzeel, 2008; Lutz et al., 2014; Rees and Collins, 2006). In the eastern part of the IGB, where monsoon systems are more dominant, the monsoon precipitation is the largest constituent of the total annual surface flow (Immerzeel, 2008). It is expected that due to projected rises in temperature and precipitation changes, glaciers and seasonal snow cover will be affected, eventually affecting the amount of meltwater and thus the amount of surface water supply from upstream mountainous basins, especially in the western part of the IGB (Kraaijenbrink et al., 2017; Viste and Sorteberg, 2015). Further, monsoon dynamics will likely change, resulting in a decreasing number of rainy days, increasing intensity of precipitation, and increasing mean monsoon precipitation (Kumar et al., 2011; Lutz et al., in review; Sharmila et al., 2015; Turner and Annamalai, 2012). This might eventually affect the water supply patterns in the eastern part of the IGB. On top of that, long-term precipitation changes may lead to changes in groundwater recharge and storage, which in turn will affect groundwater availability (Asoka et al., 2017). There are however large uncertainties in the projected precipitation changes due to the large spread among the different climate model runs (Arnell and Lloyd-Hughes, 2014; Lutz et al., 2016b; Moors et al., 2011), which hampers the projection of future water supply rates. In addition to climate-induced changes in surface and groundwater supply, groundwater depletion is expected to intensify over the next decades due to socio-economic development (Rodell et al., 2009; Wada, 2016; Wada et al., 2010).

Simultaneous with changes in water supply under climate change, rapid and continuous population growth and strong economic development are expected to result in a rapid increase in water demand over the coming decades (Biemans et al., 2011; Rasul, 2014, 2016; Wada et al., 2016). The population in the IGB is expected to grow from 900 million inhabitants in 2010 to 1.1 - 1.4 billion inhabitants in 2050, which will likely be accompanied with rapid urbanization (Klein Goldewijk et al., 2010; Rasul, 2016). For instance, in countries like India and Pakistan, the expectation is that by 2050 more than 50% of the population will live in urban areas (Mukherji et al., 2018; UN-DESA, 2018). The population growth is also expected to be accompanied with continuing fast economic growth (i.e. currently between 2.5% and 7.3% per year (ADB, 2018)), rapid industrialization, and an intensification of water use in food production (e.g. due to expansion of irrigated areas) (Biemans et

al., 2013; Rasul, 2016). This will likely result in an increasing pressure on water resources, which in turn will affect food security, safe access to drinking water, public health, and environmental well-being (Liu et al., 2017; Taylor, 2009). Over the past decades, blue water scarcity has already become a prominent issue in some parts of the IGB. Hoekstra et al. (2012) found, for instance, that the Indus river basin experiences severe blue water scarcity during eight months a year, whereas the lower parts of the Ganges-Brahmaputra river system face severe blue water scarcity during five months a year.

Towards the future, it is rather uncertain how blue water scarcity will develop in the IGB. Some (global) studies (e.g. Alcamo et al., 2007; Arnell, 2004) found that future blue water scarcity will decline, mainly by increasing water availability due to climate change, whereas other studies (e.g. Gain and Wada, 2014; Hanasaki et al., 2013; Vörösmarty et al., 2000) found that future (seasonal) blue water scarcity will increase due to socio-economic changes, mainly resulting from population growth, or due to decreasing water availability. The opposing trends in future blue water scarcity found in the cited studies indicate that the uncertainty in how the future South Asian blue water gap will develop is large and that an improved understanding on the development of the regional blue water gap is needed. One of the drawbacks in the cited studies is, for example, that, in general, the selection of climate models, RCPs, and SSPs (RCP = Representative Concentration Pathway; SSP = Shared Socio-economic Pathway) was not tailored to the representation of a wide range of possible futures in terms of climate change and socio-economic development. Consequently, the projected blue water scarcity trends may not provide a full picture in how blue water scarcity will develop into the future. Model selection approaches (e.g. Lutz et al., 2016b) with a focus on a wide range of possible futures in terms of climate change, and the selection of contrasting RCP-SSP combinations according to a RCP-SSP Framework (van Vuuren et al., 2014), can for instance be used to eliminate this drawback. Another drawback is that no models were used with a sufficient representation of cryospheric-hydrological processes. Therefore, the lack of proper simulations of the evolution of mountain water resources (e.g. glacier evolution) may have imposed uncertainties in the outcomes of these studies. Models with a sufficient representation of cryospheric-hydrological processes can be used to eliminate this drawback.

(Blue) water scarcity has been assessed by different methodologies over the last decades. One type of assessments relied on statistics of water use (e.g. FAO AQUASTAT) and observations of meteorological and hydrological variables (Bierkens, 2015). Other were conducted by using several model types, such as global hydrological models (e.g. H08 (Hanasaki et al., 2008a, 2008b), LPJmL (Schewe et al., 2014) and PCR-GLOBWB (Van Beek et al., 2011; Wada et al., 2014)) (Veldkamp et al., 2017). There are several advantages of the use of hydrological models above the use of statistics. First, blue water scarcity can be assessed by taking water availability, the main types of water use (i.e. agricultural, domestic, and industrial), and their relationships and feedbacks into account on a high spatial and temporal resolution (e.g. 5 arc min and daily). Second, models such as the LPJmL model can be used to assess the impacts of human interventions (e.g. reservoirs) on water availability and irrigation water supply (Biemans et al., 2011; Haddeland et al., 2014). Finally, the use of models contributes to an improved understanding on processes that are relevant in the development of (future) blue water scarcity.

Large scale hydrological models have mostly been applied without making an explicit distinction between up- and downstream domains and their roles in water supply and demand. To make an explicit distinction between the dominant

processes in the different domains, different tools are required to simulate the domain-specific processes properly. For instance, in the upstream domains of the IGB, water availability is highly depending on natural factors, such as ice and snowmelt (e.g. Lutz et al., 2014)). Since cryospheric and hydrological processes vary strongly over short distances in the upstream mountainous areas, higher resolution models with a robust representation of mountain-specific cryospheric and

hydrological processes are required to simulate water availability and supply in and from the upstream (mountainous) domains accurately. In the downstream domains of the IGB, the human influence on the hydrological cycle is large with large irrigation canal systems and reservoirs (e.g. Tarbela Dam) (Biemans et al., 2013). In addition, agricultural water use is a very important topic in this region, which requires knowledge of related processes, such as crop growth, and relations between water availability and food production. In these domains, therefore, a high-resolution model is required that a) has

an explicit representation of human interventions in the hydrological cycle, and b) can link hydrological processes with crop processes.

Environmental flow requirements (EFRs) have not been considered in most future water scarcity assessments. EFRs have so far only been applied by Hanasaki et al. (2013) by using an EFR module (i.e. part of the H08 model) that controls the consumptive amount of water that is withdrawn from river systems. This allows the prioritization of maintaining EFRs,

but also has the consequence that agricultural production might be affected. According to Jägermeyr et al. (2017) up to ~30% of the agricultural production in South Asia can be lost when EFRs are considered. In the IGB, rapid and continuous population growth is expected, which will most likely be accompanied with an increase in food demand and thus requires a higher agricultural production (Biemans et al., 2013). Therefore, agricultural needs will probably be prioritized at the cost of environmental flows and water use will most likely intensify, which subsequently might alter flow regimes and the

ecological health of a river system (Döll et al., 2009; Pastor et al., 2014). To understand the impact of blue water consumption on environmental flow transgressions, it is therefore needed to estimate EFRs and to assess whether (future) EFRs are met or not.

Most studies that have assessed future blue water scarcity have only focussed on the interannual variability without focussing on the intra-annual variability. This can be considered as a disadvantage in regions with dry and wet seasons, such

as the IGB. For instance, Gain and Wada, (2014) found that, based on annual projections, future blue water scarcity is projected to be absent in the Brahmaputra river basin over the next decades. Seasonal and monthly projections indicated however that during the dry season blue water scarcity will become more severe in the future (Gain and Wada, 2014). For this reason, it is important to include the intra-annual variability in blue water scarcity projections in areas like the IGB.

The main objective of this study is to assess the combined impacts of climate change and socio-economic development

on the future "blue" water gap for the downstream floodplains of the Indus, Ganges, and Brahmaputra (IGB) river basins until the end of the 21$^{st}$ century. For the upstream mountainous domains, we apply a distributed model with a strong representation of cryospheric-hydrological processes that explicitly simulates cryospheric changes (i.e. glacier and snow cover) under climate change. For the downstream domains, we apply a distributed hydrology and crop production model with an explicit representation of human interventions in the hydrological cycle to simulate downstream water supply and

demand. We use the RCP – SSP framework to include a wide range of possible futures in terms of climate change and socio-economic development (van Vuuren et al., 2014). Both models are forced with outputs of eight downscaled General Circulation Models (GCMs) representing a region-specific wide range of possible climate conditions (i.e. representing RCP4.5 and RCP8.5) (Lutz et al., 2016b). In addition, we use a set of regional land use scenarios and socio-economic

scenarios (derived from SSP1 and SSP3 (Riahi et al., 2017)) to force the hydrology and crop production model. Water demand and consumption are estimated in terms of the amount of water that is required for withdrawal and that is consumed, respectively, by the agricultural, domestic, and industrial sectors. The blue water gap is estimated as the amount of unsustainable groundwater that is withdrawn to fulfil the blue water demand. Finally, EFRs are estimated according to the Variable Monthly Flow (VMF) method (Pastor et al., 2014) to assess the impact of (future) blue water consumption on

environmental flow transgressions, assuming that meeting EFRs have the lowest priority.

This study stands out in comparison with previous work in the region (e.g. (Gain and Wada, 2014) by means of multiple novelties. First, the novelty of this study lies in the application of a coupled modelling approach, including a high-resolution cryospheric-hydrological model (5 x 5 km) and a high-resolution hydrology and crop production model (5 x 5 arc min), that can simulate up- and downstream water availability, and the downstream water supply, demand, and gap in the entire IGB.

Second, the hydrology and crop production model applied for downstream domains, has specially been developed for this region in that it is able to a) simulate water distribution through extensive irrigation canal systems of the Indus and Ganges river basins, b) make improved simulations of the timing of water demand for agriculture due to an explicit representation of a multiple cropping system (Biemans et al., 2016), and c) simulate groundwater withdrawal and depletion rates. Third, the high-resolution models are forced with an ensemble of downscaled and bias-corrected GCMs that were selected by using an

advanced selection approach and represent a wide range of possible futures in terms of climate change for RCP4.5 and RCP8.5. Fourth, the hydrology and crop production model is forced with a set of socio-economic and land use scenarios that are most likely linked with the RCPs (i.e. according to the RCP-SSP framework). Finally, the outcomes of the hydrology and crop production model are used to assess the impact of (future) blue water consumption on environmental flow transgressions.

**2. Study Area**

The future blue water gap is examined for three major South Asian river basins, which are considered as a "hotspot" of climate and socio-economic changes: the Indus, Ganges, and Brahmaputra (De Souza et al., 2015) (Figure 1). The Indus, Ganges, and Brahmaputra river basins are selected as study area because these South Asian river basins depend to varying degrees on water generated in the Hindu-Kush Himalayan mountain ranges and at the same time have contrasting differences

in terms of hydro-climatic and socio-economic characteristics. In a geopolitically complex region, the Indus (I), Ganges (G), and Brahmaputra (B) drain surface areas of around 1,116,000 km$^2$, 1,001,000 km$^2$, and 528,000 km$^2$, respectively, and traverse Afghanistan (I), Pakistan (I), India (I, G, B), China (I, G, B), Nepal (G), Bhutan (B), and Bangladesh (G, B). In this

study, the IGB river system is subdivided in several upstream and downstream domains: the Upper Indus Basin (UIB), Upper Ganges Basin (UGB), Upper Brahmaputra Basin (UBB), Lower Indus Basin (LIB), Lower Ganges Basin (LGB), and Lower Brahmaputra Basin (LBB). Thereby, the upstream domains are dominated by the mountainous terrains of the Tibetan Plateau and Hindu Kush – Himalayan (HKH) mountain ranges with elevations up to 8850 m above sea level, and the

downstream domains are dominated by hilly regions and floodplains that are part of the Indo-Gangetic plains. The boundary between upstream and downstream domains is located at the southern margins of the Himalayan foothills and directly upstream of large reservoirs, such as the Tarbela and Mangla Dam reservoirs.

The Ganges river basin is the most densely populated basin with a population density of about 580 inhabitants/km$^2$, and the Brahmaputra river basin is the least populated basin with 131 inhabitants/km$^2$ (2016; Klein Goldewijk et al., 2010). India

has the largest economy with a nominal GDP per capita of 1604 US\$ yr$^{-1}$, whereas Nepal has the smallest economy with a nominal GDP per capita of 748 US\$ yr$^{-1}$ (International Monetary Fund, 2016). Water withdrawal (i.e. in South Asia) is highest in the agricultural sector (91%, corresponding with 913 km$^3$/year), followed by the domestic (7%, corresponding with 70 km$^3$/year) and industrial sectors (2%, corresponding with 20 km$^3$/year) (FAO, 2012). Much of the water withdrawn is used for the irrigated agricultural areas that are present in the IGB. Among the three river basins, the Ganges river basin

has the largest irrigated area with 257,000 km$^2$ (i.e. situation in 2000), followed by the Indus river basin (213,000 km$^2$) and the Brahmaputra river basin (27,000 km$^2$) (Biemans et al., 2013). In the irrigated areas of the Indus and Ganges river basins, mainly cash crops, such as sugarcane, wheat, and rice are cultivated (FAO, 2012). Thereby, the annual production of sugarcane is highest with 431 Mt, followed by rice (233 Mt), and wheat (138 Mt) (2016; FAO, 2017).

The climate of the IGB river systems is mainly dominated by the East-Asian and Indian monsoon systems, and the

Westerlies. Westerlies are most dominant in the western part of the IGB with significant precipitation during the winter period. The East-Asian and Indian monsoon systems become increasingly dominant when moving eastward causing most of the precipitation to occur during the monsoon season (June-September). In the Brahmaputra river basin, where the climate is mainly driven by the monsoon systems, 60-70% of the annual precipitation occurs during the monsoon season (Immerzeel, 2008). Annual precipitation amounts vary from less than 200 mm in the Thar desert (LIB) and the Tibetan Plateau (UIB) to

more than 5000 mm in the floodplains of the LBB (Lutz et al., in review). The high-altitude regions of the HKH experience a cold climate with annual average temperatures down to -19 ℃ in the Karakoram (UIB), whereas the downstream domains experience mild winters and hot summers with annual average temperatures up to 28 ℃ at the southern margins of the LGB (Cheema and Bastiaanssen, 2010; Lutz et al., in review; Wijngaard et al., 2017). Within the IGB two growing seasons are prevailing: the rabi season (November – April) and the kharif season (May – October) (Cheema et al., 2014; Portmann et al.,

30    2010).

## 3. Data and Methods

### 3.1 Definitions

Throughout this study, we use several terms, which we define as follows:

- Blue water: water that is withdrawn from surface water and groundwater bodies. Surface water is defined as water withdrawn directly from rivers, lakes, and reservoirs. Groundwater is defined as water withdrawn from both shallow and deep aquifers, using (artificial) wells.

- Green water: water that is infiltrated into soils and that originated directly from precipitation.

- Blue water availability: the total amount of water available in rivers, reservoirs, and groundwater.

- Blue water demand: the total amount of blue water that is required for withdrawal by the agricultural, domestic, and industrial sectors.

- Blue water consumption: the total amount of blue water that is consumed (evapotranspiration in agriculture) by the agricultural (evapotranspiration), domestic, and industrial sectors (withdrawal minus return flows).

- Blue water gap: the amount of unsustainable groundwater that is withdrawn to fulfil the blue water demand. The blue water gap occurs when the mean annual groundwater withdrawal exceeds the mean annual groundwater recharge.

### 3.2 Modelling Framework

We use a coupled modelling approach to simulate upstream water availability and downstream water supply and demand. To this end, two physically-based fully-distributed models are used: the cryospheric-hydrological Spatial Processes in HYdrology (SPHY) model (Terink et al., 2015) and an adjusted version of the (eco-)hydrological Lund-Potsdam-Jena-managed-Land (LPJmL) model (Biemans et al., 2013, 2016; Bondeau et al., 2007; Rost et al., 2008). SPHY and LPJmL are set up for a reference period (1981-2010) and a future period (2011-2100), and both run at a daily time step.

#### 3.2.1 Upstream: SPHY

We use SPHY to simulate water availability from the upstream mountainous domains of the IGB. The SPHY model is developed specifically for the high mountain environment in Asia. The model runs at a spatial resolution of 5 x 5 km and reports on a daily time step. SPHY has been used to assess climate change impacts for high mountain hydrology in Asia before (Lutz et al., 2014, 2016a; Wijngaard et al., 2017). The used set up was calibrated and validated using IceSat glacier mass balance data (Kääb et al., 2012), MODIS snow cover data (Hall et al., 2002; Hall and Riggs, 2015) and observed discharge in a study on the impacts of climate change on hydrological extremes in the upstream domains of the IGB (Wijngaard et al., 2017). The model simulates daily discharge by calculating the amount of total runoff for each grid cell, and subsequently by routing the total runoff downstream by means of a simplified routing scheme that requires a digital

elevation model (DEM) and a recession coefficient. Thereby, the total runoff is the sum of glacier runoff, snow runoff, surface runoff, lateral flow, and baseflow.

For the estimation of the contribution of glacier runoff, sub-grid variability (i.e. 1 km$^2$) is applied by determining the fractional ice cover in each cell, where fractional ice cover can range between 0 (no ice cover) and 1 (complete ice cover).

Changes in fractional ice cover over time are modelled using an approach that considers mass conservation and ice-redistribution (Terink et al., 2017). In addition to the determination of fractional ice cover, other information, such as initial ice thickness and the type of glacier (i.e. debris-free or debris-covered) is attributed to a unique identifier that is created for (a part of) each glacier within a model cell. The degree-day approach of *Hock* (2003) is used to simulate glacier melt, which is subsequently subdivided over the surface runoff and baseflow pathways by a calibrated glacier runoff fraction.

Those parts that are not covered by glaciers are covered by snow, bare soil, vegetation, or open water. For the snow-covered parts, the model of *Kokkonen et al.* (2006) is used to simulate snow storage dynamics. Snow accumulation and –melt is simulated by the degree-day approach of *Hock* (2003), whereas snow sublimation is estimated by a simple elevation-dependent potential sublimation function (Lutz et al., 2016a). Besides snow melt, accumulation, and sublimation, refreezing of snowmelt and rain are included as well. Rainfall runoff processes are simulated for those parts that are free of snow. Rain

is subdivided over two pathways: i) a direct transport to the river network by surface runoff, or ii) an indirect transport to the river network via lateral flow or baseflow. For the simulation of soil water processes, processes as evapotranspiration, infiltration, and percolation are included. These processes are simulated for a topsoil and subsoil layer. For a more detailed description of SPHY we refer to *Terink et al.* (2015).

### 3.2.2   Downstream: LPJmL

The outflows of upstream domains that are simulated by SPHY are input to the hydrology and crop production model LPJmL, where water is withdrawn by users or continues its way downstream towards the Arabian Sea or the Bay of Bengal. LPJmL has an explicit representation of human interventions in the hydrological cycle that are relevant in the downstream domain, such as dynamic calculations of irrigation demand, withdrawal and supply (Rost et al., 2008), and the operation of large reservoirs (Biemans et al., 2011). LPJmL has been applied to South Asia before (Biemans et al., 2013), but has recently

been updated to represent the agricultural practice of multiple cropping with monsoon-dependent sowing dates (Biemans et al., 2016) and the distinction between different irrigation systems (Jägermeyr et al., 2015). The LPJmL model has been tested and validated for global applications, such as river discharge (Biemans et al., 2009), irrigation requirements (Rost et al., 2008), crop yields (Fader et al., 2010), and sowing dates (Waha et al., 2012). On regional level, irrigation water withdrawals have been validated for India and Pakistan (Biemans et al., 2013, 2016). In this study, the model was further

improved to represent groundwater withdrawal and depletion and the distribution of irrigation water through the extensive canal systems in the Indus and Ganges basins. Moreover, the resolution was increased to 5 x 5 arc-min.

LPJmL simulates daily discharge by i) calculating the total amount of runoff generated for each grid cell as the sum of surface runoff, subsurface runoff, and baseflow, and ii) routing the total runoff downstream along a river network. Water

enters a grid cell by precipitation and/or irrigation water and can be subdivided over two pathways: direct transport to the river network by surface runoff and indirect transport via infiltration and subsurface runoff or baseflow (Schaphoff et al., 2017). Groundwater reservoirs are recharged from the bottom soil layers. Water can be withdrawn from the groundwater reservoirs directly, or they contribute to baseflow through a delayed outflow parameterized by a linear reservoir model.

Water can be removed from the grid cell by soil evaporation, plant transpiration, canopy interception, and percolation. Water can also be removed from the river network by lake or canal evaporation. For a more detailed description of LPJmL we refer to Rost et al. (2008) and Schaphoff et al. (2017).

In LPJmL, the daily irrigation water consumption is calculated for each grid cell as the minimum amount of additional water needed to fill the upper two soil layers to field capacity and the amount needed to fulfil the atmospheric evaporative

demand (Rost et al., 2008). The gross irrigation demand (i.e. withdrawal) depends on the soil and the type of irrigation system that is installed. We assume that all irrigated areas in the IGB rely on flood irrigation (AQUASTAT; FAO, 2014) which is less efficient than sprinkler or drip irrigation systems (Jägermeyr et al., 2015). Daily water demand for other users (i.e. households and industry) is assumed to be constant throughout the year.

Water for irrigation and other users can be withdrawn from surface water in a grid cell, surface water from a

neighbouring grid cell or a canal system (i.e. if connected), an upstream reservoir build for water supply (i.e. if in place), and groundwater bodies, respectively. If long-term groundwater withdrawals exceed long-term groundwater recharge, the withdrawal is defined as unsustainable. In this study, we define the blue water gap as the mean annual groundwater depletion rate. Not all water that is withdrawn is consumed. Water can be lost during conveyance, by open water evaporation or as a return flow into the river network. After application to the field, again only part of the water will be used for

evapotranspiration (blue water consumption), and the remaining part will recharge groundwater or discharge as return flow to the river.

## 3.3 Data

SPHY and LPJmL are forced with daily air temperature and precipitation fields from a dataset that is developed for the Indus, Ganges, and Brahmaputra river basins (Lutz and Immerzeel, 2015), which accounts for the underestimate of high

altitude precipitation, which is common for gridded meteorological forcing datasets in the region (Immerzeel et al., 2015). The datasets are based on the Watch Forcing ERA-Interim (WFDEI) dataset (Weedon et al., 2014), and are bias-corrected and downscaled from a resolution of 0.5º x 0.5º to a resolution of 5 x 5 km and 10 x 10 km for the upstream and downstream domains, respectively. The LPJmL model is also forced with downward longwave and shortwave radiation, besides daily air temperature and precipitation fields. Downward shortwave radiation is not bias-corrected, since these datasets are corrected

to observed cloud cover and by means of corrections for aerosol loadings (Weedon et al., 2010, 2011, 2014). For the application of the meteorological forcings in LPJmL the datasets were resampled to a resolution of 5 arc-min.

We use the 15-arc-second void-filled and hydrologically conditioned HydroSHEDS DEM (Lehner et al., 2008). For the use of the DEMs in SPHY the DEMs are resampled to 5 x 5 km. LPJmL uses the stream network from HydroSHEDS at 5 x

5 arc-min. Land use information in SPHY is extracted from the MERIS Globcover product (Defourny et al., 2007). In LPJmL, gridded crop fractions of 13 rainfed and irrigated crop classes for the 2 cropping seasons were derived from the MIRCA2000 dataset (Biemans et al., 2016; Portmann et al., 2010). For SPHY, soil information from the HiHydroSoil database (De Boer, 2016), which is a dataset of soil hydraulic properties derived from the Harmonized World Soil Database (FAO/IIASA/ISRIC/ISSCAS/JRC, 2012) using pedotransfer functions (Sarmadian and Keshavarzi, 2010). LPJmL soil classes were derived from the HWSD (Schaphoff et al., 2013).

Current 5-arc-min domestic and industrial water demand datasets are extracted from the PCR-GLOBWB model. In these datasets, water demands were estimated based on methods developed by (Wada et al., 2011b, 2014). Domestic water withdrawals were derived by combining decadal and yearly population data (i.e. extracted from the HYDE v3.2. database (Klein Goldewijk et al., 2010) and the FAOSTAT database, respectively), country-specific per capita domestic withdrawal data (i.e. extracted from the FAO AQUASTAT database), and water use intensities. The water use intensities take country-specific economic and technological developments into account (Wada et al., 2011b). Hence, economic developments are based on changes in GDP, electricity production, energy and household consumption. Technological developments are derived as the energy consumption per unit electricity production and accounts for domestic/industrial restructuring or improved water use efficiency (Wada et al., 2011b). Water use intensities are also used to derive industrial water withdrawal. Industrial water demands are assumed to remain constant throughout the year, whereas domestic water demands are assumed to vary throughout the year, which is depending on air temperature (Wada et al., 2010, 2011a). Not all the water that is withdrawn is consumed. A part of the water withdrawn for domestic and industrial purposes returns to the river network as return flows. The amount of return flow is calculated by means of recycling ratios that is depending on the country-specific GDP and level of economic development (Wada et al., 2011a).

### 3.4 Future Climate and Socio-economic Development

To evaluate future changes in the water supply, demand, and gap due to climate change combined with socio-economic developments we use the RCP-SSP Framework (van Vuuren et al., 2014). We force SPHY and LPJmL with an ensemble of downscaled General Circulation Model (GCM) runs from the medium stabilization scenario RCP4.5 and the very high baseline emission scenario RCP8.5 (van Vuuren et al., 2011). From the CMIP5 multi-model ensemble (Taylor et al., 2012) we select four GCM runs for each RCP that represent the full CMIP5 ensemble in terms of projected ranges in the means and extremes of future air temperature and precipitation over the IGB region, and have sufficient skill to simulate historical climate conditions in the IGB (Lutz et al., 2016b). Subsequently, the selected models are downscaled using the reference climate data by applying a Quantile Mapping approach, which performs well in downscaling climate model data for floodplains as well as mountainous terrains (Themeßl et al., 2011). This method scales future GCMs down and bias-corrects them by means of empirical cumulative density functions that are calculated for the reference climate dataset and historical GCM runs (1981-2010).

For the representation of future socio-economic development, we select two SSP storylines (O'Neill et al., 2014, 2015; Riahi et al., 2017) that represent a "Sustainability" scenario (SSP1) and a "Fragmentation" scenario (SSP3). We choose to select SSP1 and SSP3, because these SSPs are most likely linked with RCP4.5 (i.e. RCP4.5 – SSP1) and RCP8.5 (i.e. RCP8.5 – SSP3) (van Vuuren and Carter, 2014). Future 5-arc-min domestic and industrial water demand datasets are extracted from the IMAGE v3.0 model (Stehfest et al., 2014). Within the IMAGE model a sub-model (i.e. developed by Bijl et al., (2016)) is included, which calculates the future domestic and industrial water demands based on projections for population growth and economic development (based on GDP per capita) that are consistent with the selected SSPs. The projected population and GDP (PPP) changes for the IGB are summarized in Table 1 for SSP1 and SSP3.

Land use change scenarios that are consistent with the SSP storylines are calculated by Integrated Assessment Models like IMAGE (Stehfest et al., 2014). IMAGE calculates land use change based on a set of SSP-specific assumptions regarding dietary changes and resulting per capita food demand, the level of intensification and potential yield increase on existing cropland, and changes in import and export of commodities. We use the SSP1 and SSP3 regional scale outcomes of IMAGE (Doelman et al., 2018) to derive changes in rainfed and irrigated cropland extents for Pakistan, India, Nepal, and Bangladesh between 2010 and 2100. Subsequently, we project those changes on our gridded datasets of current kharif and rabi cropped areas to construct transient datasets of land use change in the IGB. These gridded datasets are used in combination with the climate change datasets to estimate future water requirements for irrigation. We assume that both the crop distribution and crop types remain as they are. This implies that they are not adapted when crop growth conditions become unfavourable (e.g. due to changing climate conditions). It is beyond the scope of this study to investigate the impact of climate change adaptation of agricultural practices on irrigation water requirements and related impacts on the blue water gap.

## 3.5 Analysis of Environmental Flows

To assess the impacts of (blue) water consumption on environmental flow transgressions we estimate environmental flow requirements (EFRs) according to the Variable Monthly Flow (VMF) method of Pastor et al. (2014). The VMF method is a valid method that considers intra-annual variability in streamflow and correlates well with locally calculated EFRs. The EFRs are calculated on monthly basis by using the discharge at the river outlets of the Indus, Ganges, and Brahmaputra under naturalized conditions (i.e. without withdrawals for irrigation and other users). First, the mean annual flows (MAFs) and mean monthly flows (MMFs) are calculated for the reference (1981-2010) and far-future periods (2071-2100). The MAFs and MMFs are then used to determine low flow (MMF≤0.4·MAF), high flow (MMF>0.8·MAF), and intermediate flow seasons (MMF>0.4·MAF & MMF ≤ 0.8·MAF). Based on the seasonal classification, subsequently EFRs are calculated where the EFR is set equal to 60%, 45%, and 30% of the MMF during low, intermediate and high flow seasons, respectively. Finally, the discharge impacted by anthropogenic water withdrawals (i.e. with irrigation and full access to groundwater) is compared with the EFRs to assess whether environmental flows are met or not.

## 4. Results and Discussion

### 4.1 Future Climate Change

In the IGB, both temperature and precipitation are projected to change towards the end of the 21st century. Figure 2 shows the projected annual and seasonal temperature and precipitation changes in the IGB for RCP4.5 and RCP8.5, at the end of the 21st century. On annual basis, temperature is projected to increase with 1.5 – 2.9 °C for RCP4.5 and 2.8 – 5.2 °C for RCP8.5, with respect to the reference period (1981-2010). The largest increases are projected in the western and north-western parts of the Indus river basin (i.e. in the Hindu Kush and Karakoram mountain ranges) and on the Tibetan Plateau. The large temperature increases in these regions can most likely be attributed to elevation-dependent warming, which causes a stronger warming in the high altitude upstream regions in comparison with the lower-lying downstream regions (Palazzi et al., 2016; Pepin et al., 2015). Precipitation is, in general, projected to increase with increases up to about 200% for RCP4.5 and up to about 100% for RCP8.5. Thereby, the largest increases are projected in the southernmost parts of the Indus river basin, which is a region where the amount of precipitation is relatively low (less than 300 mm/year) and thus small absolute increases can result in large relative increases. In the same region, also the range in model projections is large. Besides precipitation increases, also precipitation decreases are projected. These decreases are mainly projected to occur in the westernmost part of the Indus river basin. On seasonal basis, the projected temperature changes do not show large seasonal differences. The main difference can be found between the projections made for RCP4.5 and RCP8.5 with temperature differences up to about 2 °C between RCP4.5 and RCP8.5. The projected precipitation changes show large seasonal differences. For RCP4.5, the largest and smallest increases are, in general, projected during post-monsoon and pre-monsoon/winter, respectively. During the pre-monsoon and winter seasons even a decrease in precipitation is projected in the UIB (~-1%) and UGB (~-5%), respectively. For RCP8.5, precipitation increases are, in general, largest during post-monsoon. During pre-monsoon, also precipitation decreases are projected in the UIB (~-4%). The range in model projections is especially large during the post-monsoon and winter seasons.

### 4.2 Blue Water Availability

In the IGB, the seasonal and spatial variability of surface water availability is quite large. Figure 3 shows the seasonal surface water availability (i.e. natural runoff) for the reference period (1981-2010) in the upstream and downstream domains of the IGB as simulated by SPHY and LPJmL. The surface water availability is generally largest during the monsoon season (Figure 3c) varying from less than 100 mm/year in the floodplains of the Indus (LIB) to more than 3500 mm/year in the mountainous upstream domains of the Ganges and Brahmaputra. In these domains, the large surface water availability can mainly be attributed to the combined contributions from ice and snowmelt, and monsoon precipitation that can reach amounts over 3000 mm/year at the southern margins of the UGB and UBB (Wijngaard et al., 2017). During the winter season (see Fig 3a) the surface water availability is generally lowest with rates less than 100 mm/year in most regions of the IGB. Water availability is generally higher than 100 mm/year in the LBB and directly south of the Himalayan arc. The

higher surface water availability in these regions can likely be explained by the release of groundwater from aquifers that have been recharged during the monsoon season. A similar pattern can also be recognized for the same regions during the pre-monsoon (Figure 3b) and post-monsoon seasons (Figure 3d). During the pre-monsoon season surface water availability can reach up to about 1000-1500 mm/year in the HKH mountain ranges, which can be attributed to snowmelt.

Future water availability is expected to increase as a result of climate change. Figure 4 shows the current and future monthly surface water availability for the up- and downstream domains of the IGB under current (1981-2010), mid-future (2041-2070; MOC), and far-future (2071-2100; EOC) climate conditions. Surface water availability is projected to increase for, both, RCP4.5 and RCP8.5 in the entire IGB. Similar trends have also been found in other studies conducted in (a part of) the IGB (Immerzeel et al., 2010; Lutz et al., 2014; Masood et al., 2015; Nepal, 2016). The increases in surface water

availability are projected to be stronger during the monsoon season, which can likely be attributed to increases in monsoon precipitation (Figure 2) and increases in ice melt. The increases in melt (i.e. especially ice melt) are a likely reason that the natural runoff peaks in the upstream domains of the Ganges and Brahmaputra are projected to shift from July to August. Furthermore, increases are stronger for RCP8.5, with exception of the Indus basin, where a opposite trend can be observed. The opposite trend can mainly be attributed to the reduction in snowmelt towards the end of the 21$^{st}$ century, which is most

likely caused by the stronger temperature increases in the Indus basin (Figure 2), leading to a higher fraction of precipitation to fall as rain. The range among model runs is large, especially for RCP8.5, which indicates that uncertainty in future water availability projections is large, especially in the upstream mountainous domains. The graphs further show that, under current and future conditions, there is a clear upstream-downstream difference in the amount of water that is available in the Indus and Ganges with significant larger amounts of water available in the upstream domains In the Brahmaputra basin, the

upstream-downstream difference is smaller, which can be attributed to the East-Asian monsoon systems that have a high intensity in the floodplains of the Brahmaputra. The upstream-downstream differences in surface water availability indicate the significance of upstream water resources for the floodplains that are located downstream. In the future, it is projected that the upstream-downstream difference will be enhanced, implying that the dependency on upstream mountain water resources will increase.

**4.3 Blue Water Consumption**

Irrigation is by far the largest water consumer in the IGB. Figure 5 shows the annual and seasonal blue water consumption for irrigated croplands and the combined blue water consumption for domestic and industrial sectors. The maps indicate that the irrigation water consumption is largest in the Punjab and Haryana provinces (i.e. in northern part of the LIB/western part of the LGB) with consumption rates that reach over 600 mm/year on an annual basis. Also in the Sindh province (i.e. located

in the delta plains of the Indus) and along the Ganges river consumption rates are high. The difference in water consumption between the rabi (winter) and kharif (monsoon) seasons is limited in the Indus river basin, whereas in the Ganges and Brahmaputra river basins the water consumption during the rabi season is significantly higher at most of the croplands than during the kharif season. The seasonal differences are a result of rainfall patterns in the IGB. In the Ganges and Brahmaputra

river basins, the Indian and East-Asian monsoon systems prevail, which means that sufficient green water is available and thus (blue water) irrigation is less concentrated during the kharif season (Biemans et al., 2016). In the Indus river basin, the influence of monsoon systems is smaller, which means more irrigation is required to fulfil the crop demands. However, during the rabi seasons the amount of precipitation is limited, which means also (blue water) irrigation is required in the

Ganges and Brahmaputra river basins. In comparison to irrigation, the water consumption in the domestic and industrial sectors is almost negligible. In most areas, the consumption rates are less than 100 mm per year. Only in the larger urban areas, such as New Delhi, Islamabad, Lucknow, and Jaipur (location, Figure 1), the consumption rates can reach up to 380 mm/year.

As a result of climate change and/or socio-economic developments, blue water consumption is projected to change into

the future. Figure 6 shows the projected changes in the annual blue water consumption for irrigated croplands and other users (i.e. domestic and industrial sectors) for RCP4.5, RCP8.5, RCP4.5 – SSP1, and RCP8.5 – SSP3. Under current conditions (i.e. REF, 1981-2010), the total blue water consumption is largest in the Indus river basin with a total rate of 145 km$^3$/year, of which 138 km$^3$/year (~95%) is consumed on irrigated croplands and 7 km$^3$/year (~5%) is consumed by domestic and industrial sectors. The total blue water consumption is smallest in the Brahmaputra river basin, with a total rate of 5 km$^3$/year

of which 4 km$^3$/year (~80%) is consumed on irrigated croplands and 1 km$^3$/year (~20%) is consumed by domestic and industrial sectors. The differences in total water consumption among the basins, is that in the Indus river basin agriculture is dominated by irrigated croplands (see Fig 1d), whereas in the Brahmaputra river basin agriculture is dominated by rainfed croplands. In addition, the LIB covers a larger area than the LBB, which eventually result in larger consumption rates when aggregating the grid values within a basin. Future total water consumption is projected to change. When only considering

climate change, there will be no change in domestic and industrial water consumption. Irrigation water consumption is projected to decrease from 138 km$^3$/ 91 km$^3$/ 4 km$^3$ per year up to about 116 km$^3$/ 69 km$^3$/ 3 km$^3$ per year in the LIB/ LGB/ LBB for RCP8.5, at the end of the 21$^{st}$ century. This trend can be explained by growing seasons that become shorter for most crops due to temperature increases. The shorter growing seasons mean that less water is demanded and thus less water is consumed. In addition, precipitation is projected to increase (Figure 2), which means more green water will be available and

less (blue water) irrigation is required. When considering future climate change and socio-economic developments, an increase in the total water consumption is projected with mean relative increases up to about 36%/ 60%/ 147% per year in the LIB/ LGB/ LBB for RCP8.5 – SSP3, at the end of the 21$^{st}$ century. The increasing total water consumption can mainly be attributed to increasing domestic and industrial water consumption that emerge from population growth and economic development. Their increase ranges from 283% to 311% for RCP4.5 – SSP1 and from 586% to 715% for RCP8.5 – SSP3, at

the end of the 21$^{st}$ century, indicating that domestic and industrial water consumption will be a significant component of the South Asian future water balance. Compared to the reference period there is however a slight decrease in irrigation water consumption projected, although the decreases are smaller than those for the runs considering climate change only, which is due to the expansion of irrigated croplands under the SSPs. Only for RCP8.5 – SSP3 a slight increase in the irrigation water consumption is projected at the end of the 21$^{st}$ century.

Figure 7 shows the monthly projected changes in the total blue water consumption for RCP4.5, RCP8.5, RCP4.5 – SSP1, and RCP8.5 – SSP3. Under current climate conditions, two peaks in the total water consumption can be recognized in the Indus river basin, which coincide with the rabi and kharif crop seasons. In the Ganges and Brahmaputra river basins, the total water consumption is highest during the rabi season, but also smaller peaks can be recognized that coincide with the kharif season. Considering climate change only, the total water consumption is projected to decrease slightly throughout the entire year in the Indus river basin, with exception of the post-monsoon season, when a slight increase is projected. In the Ganges river basin, the total water consumption is projected to decrease during the second half of the rabi season, whereas during the first half of the rabi and kharif seasons the total water consumption is projected to increase slightly. These trends are also projected for the Brahmaputra river basin, with exception of the second half of the kharif season, where also a slight increase in total water consumption is projected, though the projected increases are smaller than for the first half of the kharif season. The projected increases can most likely be explained by increasing temperatures (Figure 2) that enhances the atmospheric evaporative demand. The increasing atmospheric evaporative demand result into higher crop evapotranspiration and thus higher irrigation water consumption. Because growing seasons are projected to become shorter in the IGB and precipitation is projected to increase (see Fig 2), total water consumption will eventually decrease in the second half of the rabi season, and for RCP8.5 also in second half of the kharif season. The projected increases during the second half of the kharif season in the Brahmaputra river basin can likely be explained by increasing temperatures that are smaller in the downstream domains of the Brahmaputra river basin than in other downstream domains (Figure 2). Due to the smaller temperature increases, the growing seasons show a smaller decline, and therefore the higher evapotranspiration rates emerging from temperature increases as well might outweigh the effect of shorter growing seasons, which eventually results in a slight increase in total water consumption. In the entire IGB, the water consumption for RCP8.5 is projected to be lower than for RCP4.5, which can most likely be attributed to the precipitation increases that are larger for RCP8.5, and thus cause blue water irrigation to be lower for RCP8.5 than for RCP4.5. When considering both climate change and socio-economic development, the total water consumption is projected to increase, where the largest increases are projected for RCP8.5 - SSP3. Thereby, the difference in projected increases between the mid of the 21st century (MOC) and the end of the 21st century (EOC) are especially large for RCP8.5 - SSP3, which can be explained by the extensive population growth that is projected at the end of the 21st century for SSP3 (Table 1). This eventually results in a larger increase in domestic water consumption. Further, the difference in projected increases between the RCP – SSP model runs and the reference model runs is especially large in the Brahmaputra river basin, which can be explained by the strong increases in domestic and industrial water consumption. For instance, for RCP8.5 – SSP3 a relative increase of 619% is projected in domestic and industrial water consumption at the end of the 21st century. Although the difference with projected relative increases in the Indus and Ganges river basins (i.e. 715% and 586%, respectively) is not large, the impact is however higher since the domestic and industrial sectors have a higher contribution in the total water consumption (i.e. ~20% for the reference period) in comparison with the Indus and Ganges river basins (i.e. ~5% and ~12%, respectively).

## 4.4 Blue Water Gap

Climate change is projected to have a mitigating effect on the future South Asian water gap, whereas socio-economic development is projected to have an enhancing effect on the water gap. Figure 8 shows the projected changes in the annual and seasonal blue water demand and supply for RCP4.5, RCP8.5, RCP4.5 – SSP1, and RCP8.5 – SSP3. In addition, Table 2 lists the ensemble mean and standard deviation of the projected relative changes in the annual and seasonal blue water gap for the end of the 21$^{st}$ century (i.e. EOC). Under current climate conditions, the total demand is largest in the Indus river basin with 767 km$^3$/year and smallest in the Brahmaputra river basin with 15 km$^3$/year. Most of the blue water supply consists of surface water (~67% in the Indus, and ~93% in the Brahmaputra). The other part consists of sustainable and unsustainable groundwater. The latter is defined as the blue water gap or the unmet demand, assuming that any unmet demand is covered by additional groundwater abstractions. The unmet demand is largest in the Indus river basin with 83 km$^3$/year (~11% of total demand), followed by the Ganges river basin with an unmet demand of 35 km$^3$/year (~11% of total demand) (Table 2). The simulated unmet demand in the Ganges river basin fall in range with reported historical values in other studies (Jacob et al., 2012; Richey et al., 2015; Rodell et al., 2009; Tiwari et al., 2009). The simulated unmet demand in the Indus river basin is more difficult to compare due to the limited amount of studies reporting groundwater depletion. Cheema et al., (2014) reports a groundwater depletion rate (i.e. unmet demand) of 31 km$^3$/year, which is lower than the simulated groundwater depletion rate in our study. The difference can mainly be explained by the fact that in our study the domestic and industrial sectors are also able to abstract groundwater, which consequently result in larger depletion rates. In the Brahmaputra river basin, no blue water gap is simulated, because all demands can be sustained by surface water and renewable groundwater. In the Indus river basin, the seasonal demand, supply, and gap are largest during the monsoon and melting season which coincides with the prevailing growing season, the kharif. In the Ganges and Brahmaputra river basins, the seasonal demand, supply, and gap (i.e. only in the Ganges river basin) are largest during the winter, which coincides with the rabi season. Assuming climate change without socio-economic development, demand and supply are projected to decrease in all basins on annual basis, and in general during the winter, pre-monsoon and monsoon seasons for RCP4.5 and RCP8.5. During the monsoon (i.e. only in the Brahmaputra river basin) and post-monsoon seasons, demand and supply are projected to increase. The water gap is projected to decrease under all circumstances with mean annual relative decreases up to 37% and 55% (Table 2), in the Indus and Ganges river basins, respectively, for RCP8.5, at the end of the 21$^{st}$ century. On seasonal basis, the largest relative decreases are projected during the winter season with relative decreases up to 47% (RCP4.5; EOC) and 64% (RCP8.5; EOC) (Table 2) in the Indus and Ganges river basins, respectively. The decreasing demand (met and unmet), and supply can mainly be explained by shorter growing seasons that emerge from temperature increases, and increasing precipitation that result in a shift from blue water irrigation to green water or rainfed irrigation. The increases in monsoon and post-monsoon (i.e. first half of the kharif (monsoon) and rabi (post-monsoon) seasons) can likely be explained by enhanced atmospheric evaporative demands and resulting increases in crop evapotranspiration that emerge from temperature increases. Despite the increases in demand, the water gap is projected to decrease, which can mainly be

explained by the higher surface water availability (Figure 4) that eventually result in lower unsustainable groundwater withdrawals and thus a smaller water gap. Climate change and socio-economic developments combined result, on annual base, in increasing water supply and demand in the Brahmaputra and Ganges river basins for all RCP - SSP scenarios. In the Indus river basin, only increases are projected for RCP8.5 - SSP3. For RCP4.5 - SSP1, demand and supply slightly decrease. The reason for the decreasing trend is that the (relative) increase in domestic and industrial water consumption is limited in comparison with those projected under RCP8.5 - SSP3 and other basins, which in combination with declining irrigation water demand, eventually results in decreasing water demand and supply. The future water gap tends to increase for RCP8.5 - SSP3 in the Indus and Ganges river basins with annual relative increases up to 7% and 14%, respectively, at the end of the 21$^{st}$ century (Table 2). On seasonal basis, the relative increases are largest during the monsoon season with increases up to 30% and 55% in the Indus and Ganges river basin, respectively. For RCP4.5 - SSP1 the gap decreases, since the declining irrigation water withdrawals are not outweighed by the increases in domestic and industrial water consumption. This might also explain why the water gap for RCP8.5-SSP3 is projected to decline during the winter season. Finally, the changing water demands result in changing shares of the different sectors in the total water demand, which is especially striking during the pre-monsoon season in the Brahmaputra river basin. Due to a combination of increasing domestic and industrial water demand, and declining irrigation water demand (which is especially large during pre-monsoon in this basin) the domestic and industrial sectors are eventually projected to become the largest contributors to the total water demand. The uncertainties that are accompanied with the relative changes are especially large for the RCP-SSP combinations (Table 2), which can be attributed to the large range in model outcomes that are generated for the different climate models in combination with SSP projections. For instance, the combination of RCP8.5 climate models and SSP3 projections result in an increasing water gap for some GCM-SSP combinations, whereas for others a declining water gap is projected. The combination of these changes eventually result in low ensemble means, but large standard deviations.

Figure 9 shows the spatial distribution of current groundwater depletion (i.e. indicator for the blue water gap) and future absolute changes in groundwater depletion for RCP4.5, RCP8.5, RCP4.5 – SSP1, and RCP8.5 – SSP3. Under current conditions, groundwater depletion is largest in the Punjab and Haryana provinces with depletion rates of around 1000 mm/year in the irrigated areas. In urban areas, such as New Delhi, depletion rates can even reach up to about 2000-2500 mm/year. Also in the Sindh province, the water gap is large with depletion rates in the range 300-350 mm/year. The simulated depletion rates in the irrigated areas of the Indus river basin are similar with those that were found by Cheema et al. (2014). For RCP4.5 and RCP8.5, in general less groundwater depletion is projected, which is mainly caused by the declining irrigation blue water withdrawal and consumption. For both RCP – SSP combinations, depletion is expected to decrease in the irrigated croplands, whereas in the urban areas (e.g. New Delhi) depletion is projected to increase with more than 200 mm/year (i.e. corresponding with a relative increase of more than 150%). For RCP8.5 – SSP3, also areas located in the Sindh province, and west of the Indus river are expected to experience more depletion, due to population growth and economic development.

## 4.5 Environmental Flows

The future socio-economic developments and associated increases in blue water consumption are expected to have a limited impact on environmental flow transgressions. Figure 10 shows the ensemble mean and range of the projected changes in EFRs and anthropogenic influenced discharge at the outlets of the Indus, Ganges and Brahmaputra under present and far-future (EOC) RCP-SSP conditions. Under current conditions, EFRs in the Indus, Ganges, and Brahmaputra are generally not met during the low flow season (i.e. winter, pre-monsoon, and post-monsoon), whereas during the monsoon season EFRs are met. The combination of high unmet demands in the Indus river basin (Figure 8) on the one hand and sustained EFRs on the other, can be explained by the absence of water shortage during the monsoon season due to the higher surface water availability. During the low flow season, however, the surface water availability is low, which eventually causes that EFRs and water demands cannot be met, and that high competition between different water users occur. Future projections indicate that both EFRs and anthropogenic influenced discharge will increase, which can most likely be attributed to the increase in surface water availability (Figure 4). Future EFRs are projected to be sustained during high flow seasons, whereas during low flow seasons EFRs remain unmet. However, due to low withdrawals in the Brahmaputra river basin it is projected that EFRs can be sustained all-year round. Further, the large uncertainty bands in the model projections of the Indus indicate that, especially for RCP8.5-SSP3, there is a probability that EFRs will not be met either during the second half of the monsoon season.

## 4.6 Comparison with other studies

The projected changes in the future water demand are, in general, in line with reported trends in other studies, although different processes can be responsible for the changes. In their global scale study, Wada et al., (2013) projects for instance also decreases in the irrigation water demand for RCP4.5 in the irrigated croplands of South Asia. Nevertheless, the authors project an increase in irrigation water demand for RCP8.5. According to the authors, increases in precipitation are responsible for the decrease in irrigation water demand for RCP4.5, and are outweighed by increases in temperature for RCP8.5, which cause atmospheric evaporative demand to enhance, eventually resulting in increasing irrigation water demands. In our study, the seasonal increases in irrigation water demand (i.e. during the monsoon (partly) and post-monsoon seasons) can also be attributed to enhanced atmospheric evaporative demands emerging from temperature increases. Nevertheless, other processes are responsible for the decreases in irrigation water demand. Besides increases in precipitation, shorter growing seasons as a response to temperature increases, which are larger for RCP8.5, lead to decreasing irrigation water demands. Another study of Hanasaki et al., (2013) show similar trends with decreasing irrigation water demands that are the result of increasing precipitation too. The cited study is also in line with the projected changes in water scarcity in our study with projected increases in water scarcity due to population growth and economic developments.

There are also studies that show opposite trends. For instance, Alcamo et al., (2007) show that water scarcity will decrease in South Asia due to increasing water availability that outweigh the increases in water demand. Another study of

Gain and Wada (2014), show that future water scarcity will increase in the Brahmaputra basin during the dry season (i.e. November – May), whereas in our study no water gap has been simulated or projected. The differences between the outcomes of the cited studies and those that are simulated or projected in our study is that a) different indicators were used to assess water scarcity, and b) different models and scenarios were used to assess future water scarcity. In both cited studies, the ratio between availability and consumption and or demand were used as indicator for water scarcity, whereas in our study the unsustainable groundwater withdrawal was used as an indicator for water scarcity. Since in the Brahmaputra, blue water availability is high, and blue water demand is relatively low in comparison with other basins, it means that unsustainable groundwater withdrawal is not needed to fulfil the water demands, and that therefore no water scarcity appears. Further, the use of different models and scenarios can result in different water availability projections, which can make a difference in whether water scarcity will appear or not. In our study, the increasing water availability cannot outweigh the increases in water demand, whereas this is the case in the study of Alcamo et al. (2007). The use of different water scarcity indicators and modelling approaches hampers the comparison of outcomes with those that are reported in other studies.

## 4.7  Uncertainties and Limitations

The projections of future water availability, demand, and supply are subject to several uncertainties and limitations that are mainly related to the climate change projections, the representation of (physical) processes and non-stationarity in the used hydrological models, and the land use change and socio-economic scenarios.

To assess the impacts of climate change on the future water gap, an ensemble of 8 downscaled and bias-corrected GCMs were used that cover the full range of climate conditions representative for RCP4.5 and RCP8.5. The GCMs have a poor skill in simulating the regional climate in the complex (mountainous) terrains of Central and South Asia (Lutz et al., 2016b; Seneviratne et al., 2012). Despite the selection of GCMs based on their skill in simulating the regional climate by using an advanced envelope based selection approach (Lutz et al., 2016b), still uncertainties can be introduced in the water scarcity assessments. In addition, uncertainties can be introduced in the way how GCM runs were selected. The models were selected in three consecutive steps that are based on changes in climatic means and extremes, and the skill in simulating the historical regional climate. Which method is chosen to select the climate models dictates which models are selected and therefore largely determine the outcomes of climate change impact study like ours.

There is wide variety in approaches that can be used to assess water scarcity (Liu et al., 2017). Some approaches focus only on blue water scarcity, whereas other approaches focus on the green water scarcity or the combination of blue and green water scarcity. To assess the blue and/or green water scarcity there is wide variety of indicators that can be used, where each indicator can result in a different trend. Further, the use of different models can result in different outcomes. In our study, we focus only on the blue water gap and assess blue water scarcity by using unsustainable water withdrawal as an indicator. The confidence in the trends we found by using our approach of two coupled models could be increased by including more hydrological models in a multi-model approach in combination with multiple water scarcity indicators that can increase the robustness of these trends (e.g. Alcamo et al., 2007, Wada et al., 2016).

The LPJmL model version we used for our assessments has a limitation in simulating domestic and industrial water demand. In the current version, only annual values of domestic and industrial demand could be included. Since domestic water demand varies on monthly basis with higher demands during the summer/monsoon season (i.e. higher temperatures during summer/monsoon result in higher demand) and lower demands during the winter season. This means that on seasonal base, the domestic water demand and consequently the water gap can be overestimated during the winter season, and underestimated during the summer/monsoon season. Further, the model has the limitation that the impact of water pollution on water availability cannot be simulated. This means that surface water and sustainable groundwater withdrawals can be overestimated and unsustainable groundwater withdrawals (i.e. the water gap) to be underestimated.

Whereas the LPJmL model includes human interventions, such as dam operations, and irrigation withdrawals and distribution through canals, the SPHY model that has been used for our upstream assessments does not include them. Since human interventions can influence the hydrological cycle, uncertainties might be introduced in the outflows of the upstream domains. Current impacts of dams and irrigation withdrawals are however assumed to be small due to the relative low number and total capacity of dams in the upstream domains compared to the number and total capacity of dams in the downstream domains. For instance, Tarbela Dam has a total capacity of 12 km$^3$, whereas the total capacity of dams in the upstream domains reach up to about 5.5 km$^3$ distributed over about 50 dams (FAO, 2016). Furthermore, most dams are designed as hydropower dams with limited storage or for run-off-the-river hydropower operations, which have a low degree of regulation in the upstream domains of the IGB (FAO, 2016; Lehner et al., 2011). The impact of agriculture is also assumed to be small due to the rather low irrigation water demands (and cropping intensity) in upstream domains (i.e. <100 mm yr$^{-1}$) compared to the irrigation water demands (and cropping intensity) in downstream domains (Biemans et al., 2016).

The parameterization of the SPHY and LPJmL models are based on present climatology, land use and other physical catchment characteristics, and is assumed to be stationary. Many hydrological parameters, such as parameters controlling snow processes, are however non-stationary, and can change due to possible changes in climate, land use or other characteristics (Brigode et al., 2013; Merz et al., 2011; Westra et al., 2014). According to several studies (e.g. Brigode et al., 2013; Vaze et al., 2010; Westra et al., 2014) the impact of non-stationarity is highly dependent on several factors, including the length and variability of the period of parameterization, which are decisive for the robustness of the models and thus the magnitude of uncertainty in the model outcomes. For instance, Vaze et al. (2010) indicated that models can be used for climate impact studies when parameterizations are based on data records of 20 years and longer, and for areas where future annual precipitation is not more than 15% dryer or 20% wetter than the mean annual precipitation that is derived from the data records. Other studies (e.g. Brigode et al., 2013) have indicated that shorter periods (e.g. 3 years) also can result in acceptable parameter sets. The disadvantage remains however that in the IGB long data records are scarce and future changes in climate and land use can be more extreme, especially in the southern part of the IGB, where precipitation increases over 100% are projected for the end of the 21$^{st}$ century (Figure 2). This indicates that the non-stationarity of hydrological parameters can result in uncertainties in the (future) model outcomes, such as hydrological flow predictions. To reduce the impact of non-stationarity other calibration strategies, such as the Generalized Split Sample Test procedure

(Coron et al., 2012), are recommended, which aims at testing several possible combinations of calibration-validation periods to test the model's performance under different climate conditions.

Land use change scenarios that are consistent with SSP1 and SSP3 were extracted from the IMAGE model (Doelman et al., 2018), and represent future changes in rainfed and irrigated cropland extents. One limitation is that only outcomes on future cropland extents were used as a representative for the land use change scenarios, whereas outcomes on future intensification of current croplands were not considered. Consequently, the projected yield increases and related increase in irrigation water consumption, though not linearly related, were not accounted for. This might eventually result in an underestimation of irrigation water demand. Further, future irrigation water demand can be overestimated since any future increases in irrigation efficiency were not included in our modelling approach. Another limitation that might influence the projections on irrigation water demand is the way how irrigation practices are reflected within our modelling approach. In our approach, it is assumed that crop types are not adapted over time, which consequently results in decreasing irrigation water demands when growing seasons shorten. The reality however is that farmers may adapt to changing climate conditions (e.g. due to the higher risk for heat stress that is a consequence of increased temperature (extremes)) by switching to different crop types that are more suitable for the changed climate. This might eventually influence projections on future irrigation water demand.

The SSP storylines that are used to project future changes in water demand do not account for potential feedbacks between climate change and socio-economic changes. For instance, the impacts of climate change on the land system are not included (Doelman et al., 2018). According to Nelson et al. (2014), climate change has an impact on agro-economic variables, such as agricultural area and production. The authors found, for example, that under climate change agricultural areas are projected to increase due to intensifying management practices that are induced by climate change. This means that without taking potential feedbacks between climate change and socio-economic changes into account, any future increases in cropland extents might be underestimated.

Finally, future changes in the water demand and gap that have been assessed are based on selected climate change scenarios and SSP storylines. The future changes that are assessed do however not reflect the impact of adaptation strategies. For instance, it is most likely that extra hydropower dams and reservoirs will be developed in the future (Mukherji et al., 2015). In the agricultural sector, it is most likely that irrigation efficiencies will be improved by changing irrigation systems or that crop types will be changed to ones that are more climate-tolerant (e.g. Biemans et al., (2013). Further, future developments, such as regional or transboundary cooperation that improve water and energy sharing and thus optimize water resources use (Molden et al., 2017), and its impact on the water gap have not been assessed. Follow-up studies including the simulation of basin-scale effects of climate change adaptation measures are needed to investigate the impacts of future adaptation strategies and developments on the South Asian water gap and their potential in closing the water gap.

## 5. Conclusions

The objective of this study is to assess the impacts of climate change and socio-economic developments on the future blue water gap in the downstream domains of the Indus, Ganges, and Brahmaputra river basins. To this end, we use a coupled modelling system consisting of the cryospheric-hydrological SPHY model, and the global dynamic hydrological and crop production model LPJmL. The models are forced with an ensemble of 8 bias-corrected downscaled GCMs that represent the full range of regional RCP4.5 and RCP8.5 climate conditions in combination with and without two socio-economic development scenarios (SSP1 and SSP3) that are likely linked with these RCPs. The model outcomes are analysed in terms of changes in the water availability, demand and gap.

The outcomes indicate that surface water availability will increase towards the end of the 21$^{st}$ century with the largest projected increases for RCP8.5. Thereby, increases are projected to be stronger during the monsoon season, which can mainly be attributed to the increases in monsoon precipitation and glacier melt. The upstream – downstream difference in water availability is largest in the Indus and Ganges river basins, whereas in the Brahmaputra river basin this difference is relatively small. This indicates that the dependency on upstream water resources is large, especially in the Indus and Ganges river basins. Future upstream-downstream differences in water availability are projected to be enhanced, implying that the dependency on upstream water resources will increase.

Annual and seasonal water consumption are projected to decrease when considering climate change only. This is mainly caused by shortening of growing seasons that emerge from temperature increases, and precipitation increases that result in a shift from blue water irrigation to green water or rainfed irrigation and thus cause irrigation water consumption to decline. Only in the monsoon (partly) and post-monsoon, water consumption is expected to increase, which can mainly be attributed to enhanced atmospheric evaporative demand and resulting increases in crop evapotranspiration that emerge from temperature increases. The combination of climate change and socio-economic development result in increasing annual and seasonal water consumption for RCP4.5 – SSP1 and RCP8.5 – SSP3 due to population growth and economic developments.

Due to declining water demand under climate change only, the water gap is also expected to decrease with relative decreases up to 37% and 55% in the Indus and Ganges, respectively, for RCP8.5, at the end of the 21$^{st}$ century. The combination of climate change and socio-economic development is expected to result in increasing water gaps with relative increases up to 7% and 14% in the Indus and Ganges, respectively, for RCP8.5-SSP3, at the end of the 21$^{st}$ century. Future EFRs are projected to be sustained during high flow seasons, whereas during low flow seasons EFRs cannot be met in the Indus and Ganges river basins. Based on the outcomes it can be concluded that socio-economic development is the key driver in the evolution of the South Asian water gap, whereas climate change plays a role as a decelerator. For the South Asian region, which is already facing water stress in a geopolitically complex situation, our findings provide valuable insights in the future evolution of the regional water gap, providing a scientific basis for the formulation of transboundary climate change adaptation policies.

*Code and Data Availability.* The code of the SPHY model is publicly available at https://github.com/FutureWater/SPHY. The code of the LPJmL model is available upon request. The datasets that are produced in this study are available upon request.

5 *Author Contributions.* The study is designed by A.F. Lutz, H. Biemans, R.R. Wijngaard, and W.W. Immerzeel. The model codes were adjusted by H. Biemans for the LPJmL model. The models were run by R.R. Wijngaard (SPHY) and H. Biemans (LPJmL). The analyses on the simulation outcomes were performed by R.R. Wijngaard. R.R. Wijngaard and H. Biemans prepared the manuscript text. The figures were prepared by R.R Wijngaard. Finally, the proof-reading was performed by all (co-)authors.

*Competing Interests.* The authors declare that they have no conflict of interest.

*Acknowledgements.* This work was carried out as part of the Himalayan Adaptation, Water and Resilience (HI-AWARE) consortium under the Collaborative Adaptation Research Initiative in Africa and Asia (CARIAA) with financial support 15 from the UK Government's Department for International Development and the International Development Research Centre, Ottawa, Canada. This work was also partially supported by core funds of ICIMOD contributed by the governments of Afghanistan, Australia, Austria, Bangladesh, Bhutan, China, India, Myanmar, Nepal, Norway, Pakistan, Switzerland, and the United Kingdom. This project has received funding from the European Research Council (ERC) under the European Union's Horizon 2020 research and innovation programme (grant agreement No 676819). This work is part of the research 20 programme VIDI with project number 016.161.208, which is (partly) financed by the Netherlands Organisation for Scientific Research (NWO). DFID and IDRC funds the HI-AWARE consortium, of which ICIMOD and FutureWater are consortium members. We thank Rens van Beek for supporting us in making the domestic and industrial water demands available. We thank one anonymous reviewer and the editor for their constructive remarks and suggestions that helped us to improve the manuscript significantly.

*Disclaimer.* The views expressed in this work are those of the creators and do not necessarily represent those of the UK Government's Department for International Development, the International Development Research Centre, Canada, or its Board of Governors, and are not necessarily attributable to their organizations.

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

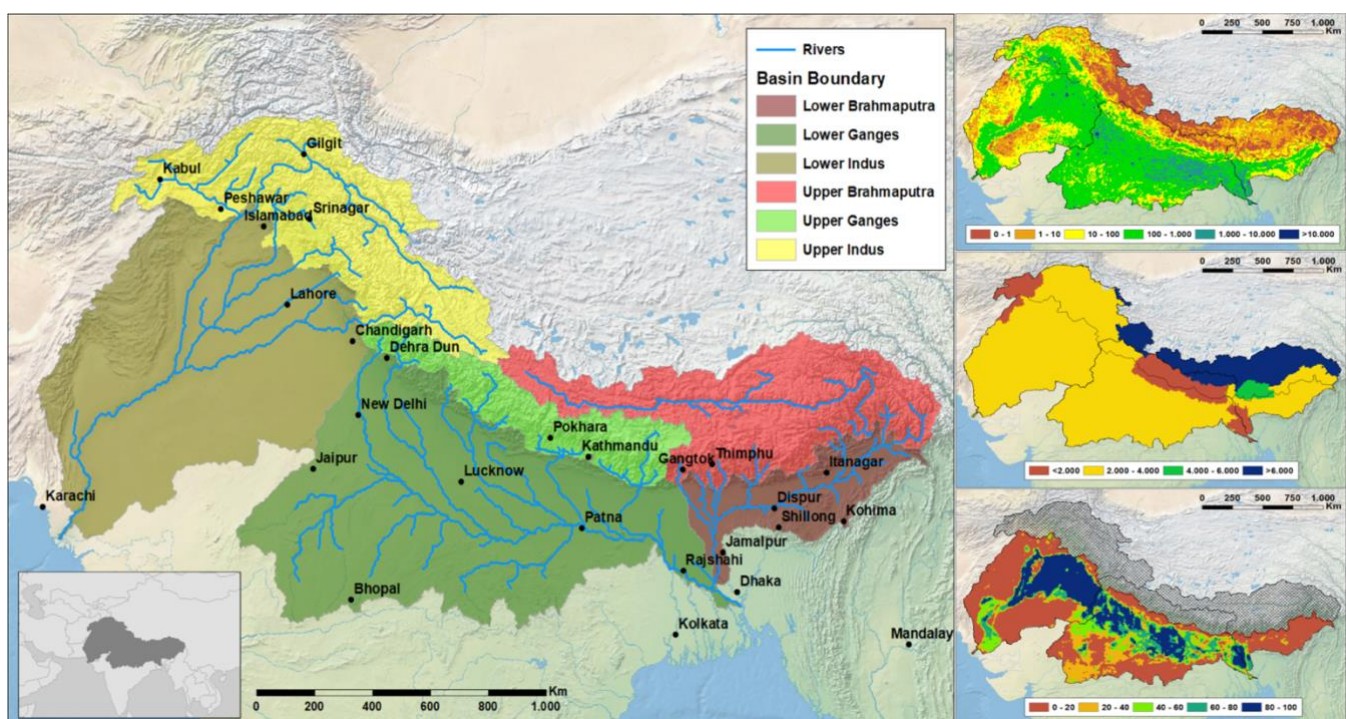

**Figure 1**. a) Map of study area showing the sub-basins and the largest cities in the region, b) the population density [inhabitants/km$^2$], c) the GDP (PPP) per capita per country [US\$/inhabitant], and d) the fraction of irrigated cropland [%]. Source of the background imagery, the cities, and the political borders illustrated in the inlet is naturalearthdata.com. Source of the population density data is the HYDE v3.2 database (Klein Goldewijk et al., 2010). The GDP (PPP) per capita is derived from IIASA SSP database (IIASA, 2017). The fraction of the irrigated cropland is derived from the MIRCA2000 dataset (Biemans et al., 2016; Portmann et al., 2010)

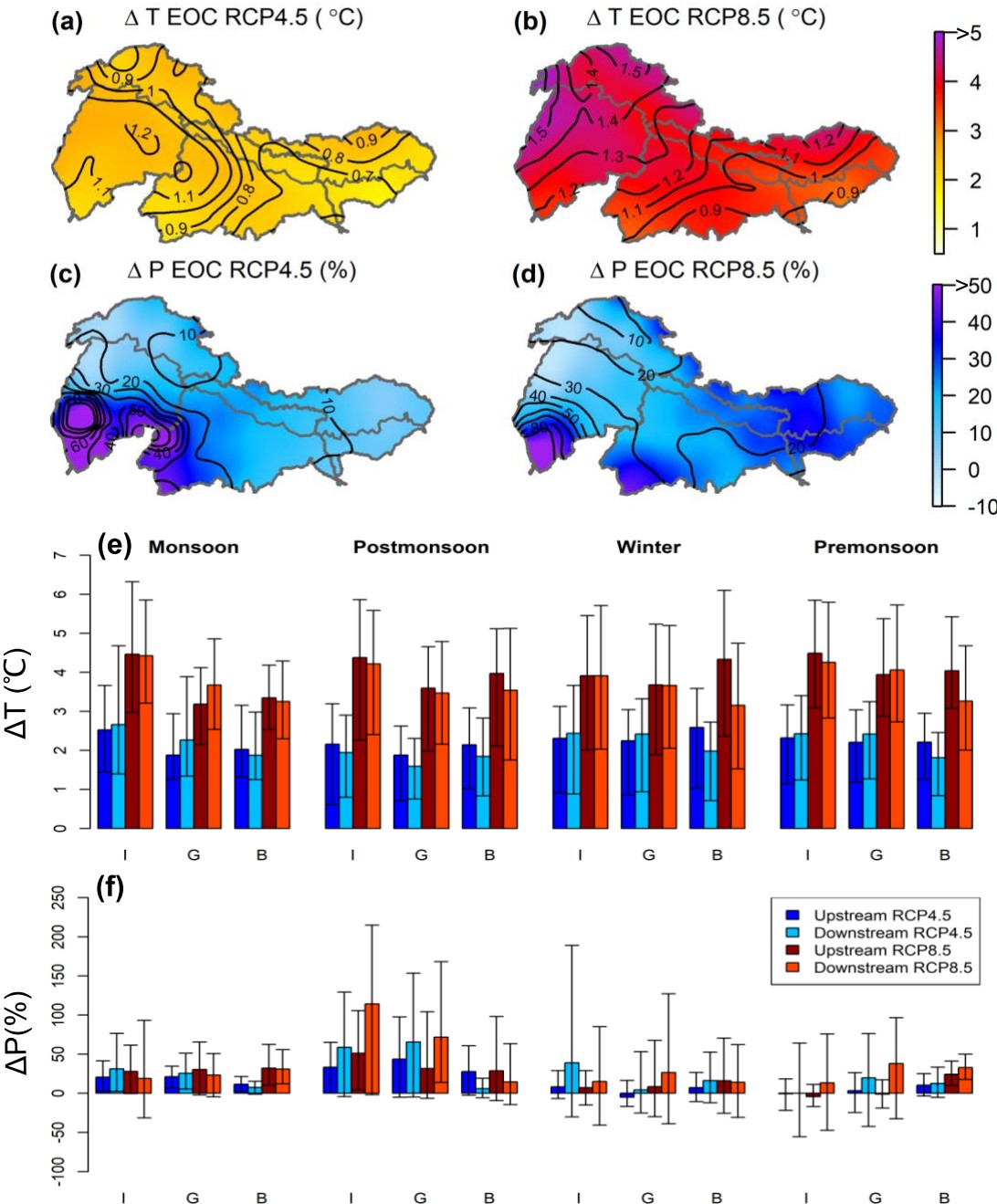

**Figure 2.** Maps showing the annual changes in temperature (a, b) and precipitation (c, d) between 2071-2100 and 1981-2010 for RCP4.5 and RCP8.5. The bar plots show seasonal changes in temperature (e) and precipitation (f) in the upstream and downstream domains of the IGB for RCP4.5 and RCP8.5. The contour lines within the maps and the error bars within the bar plots denote the ensemble range of the projections.

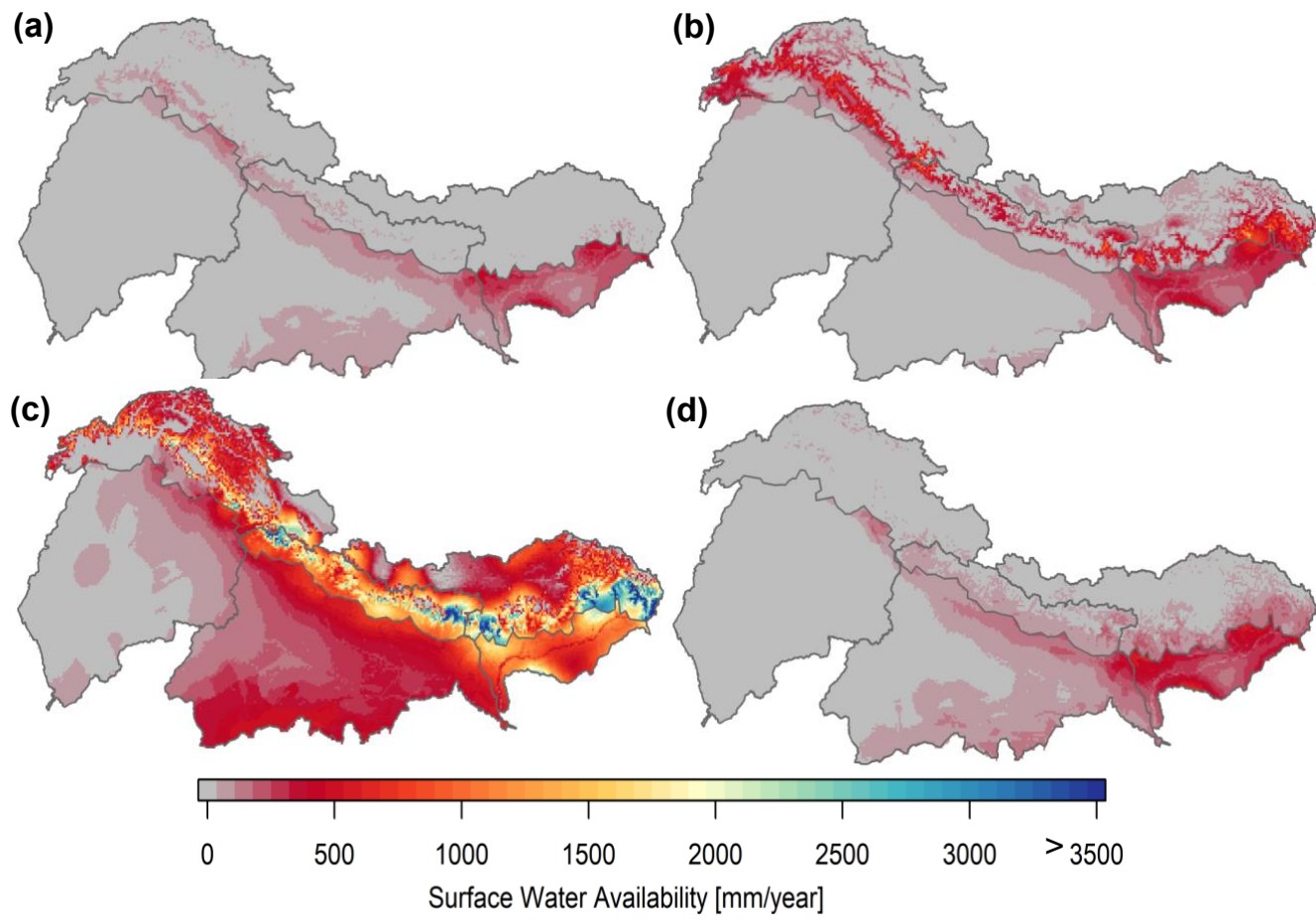

**Figure 3.** Maps showing the surface water availability in winter (a), pre-monsoon (b), monsoon (c), and post-monsoon(d).

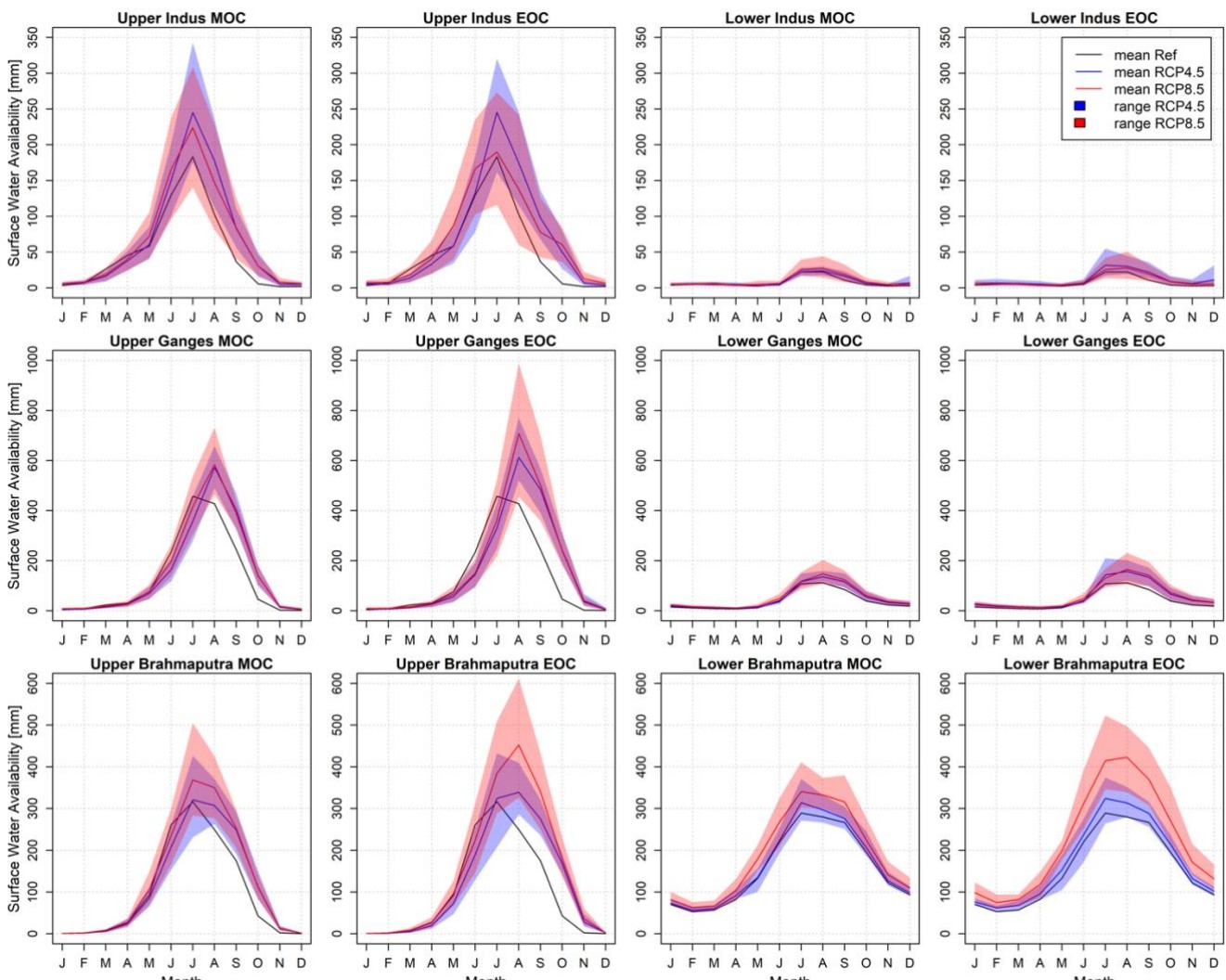

**Figure 4**. Plots showing the mean monthly blue water availability for the reference (1981-2010) and future periods (mid-of-century (MOC) (2041-2070) and end-of-century (EOC) (2071-2100)) under RCP4.5 (blue) and RCP8.5 (red). The coloured bands represent the range of ensemble projections that are resulting from forcing the SPHY and LPJmL models with the different climate models.

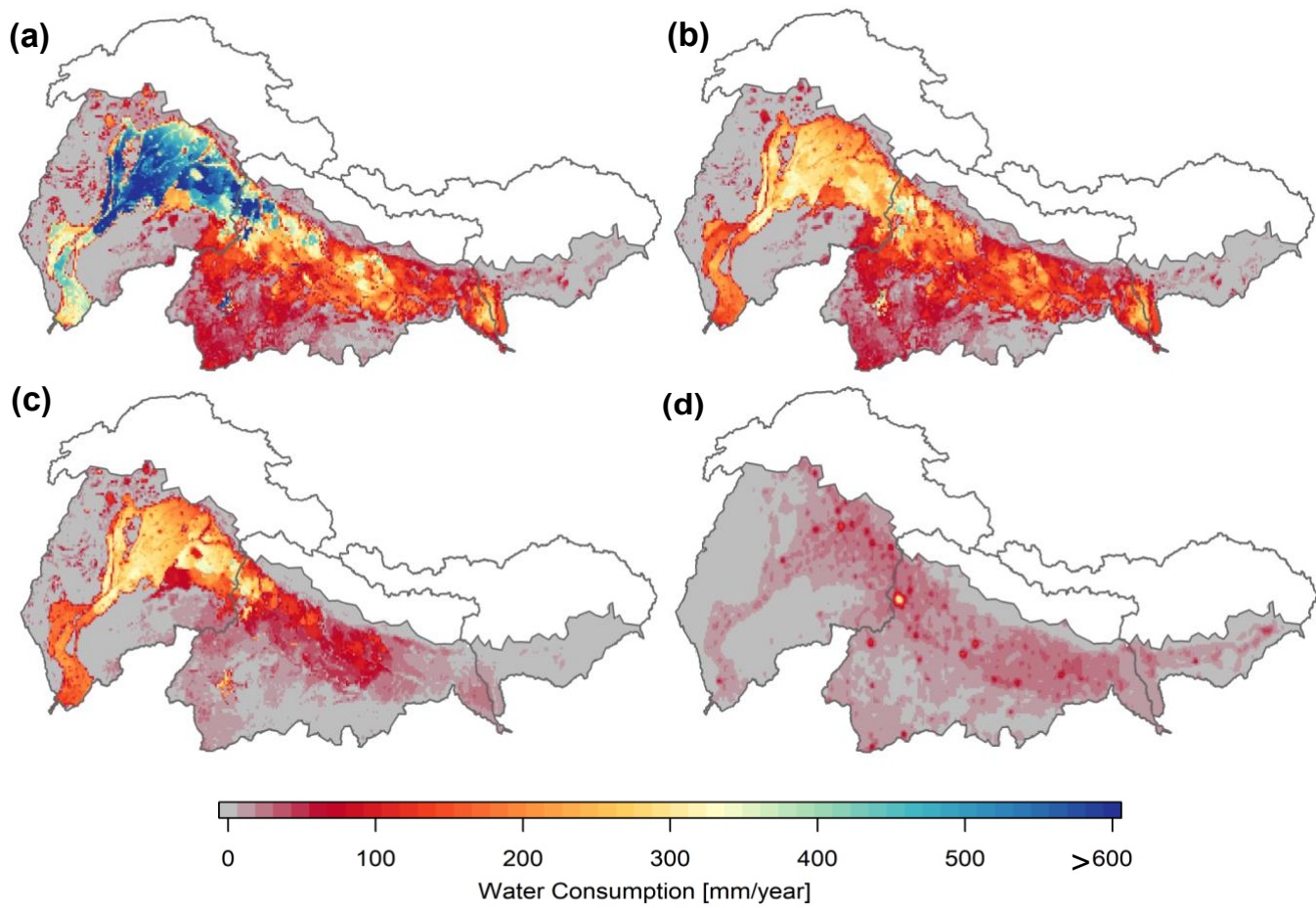

**Figure 5.** Maps showing the blue water consumption for irrigated croplands (a-c) and other users (i.e. domestic + industrial) (d). The irrigation water consumption is given on annual base (a), and for the rabi (b) and kharif seasons (c). The domestic + industrial water consumption is given on annual base.

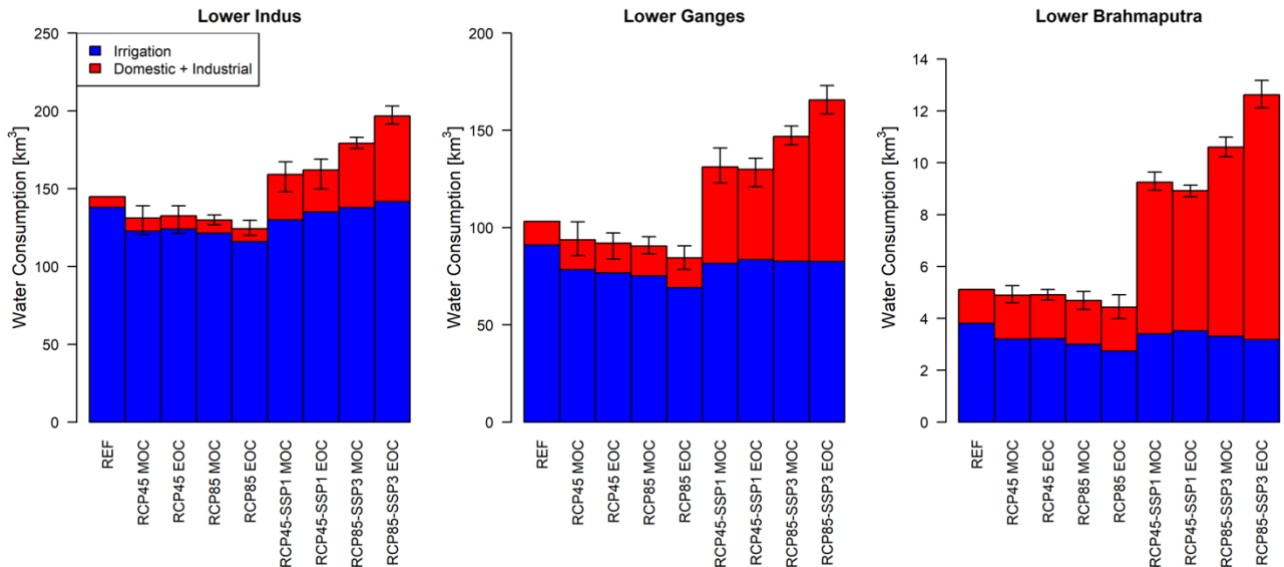

**Figure 6.** Projected changes in the annual blue water consumption for irrigated croplands and other users (i.e. domestic + industrial) for RCP4.5, RCP8.5, RCP4.5 – SSP1, and RCP8.5 – SSP3. The projected changes are given for the mid and end of the 21st century (MOC and EOC) and represent the ensemble mean. The error bars denote the range of the ensemble projections.

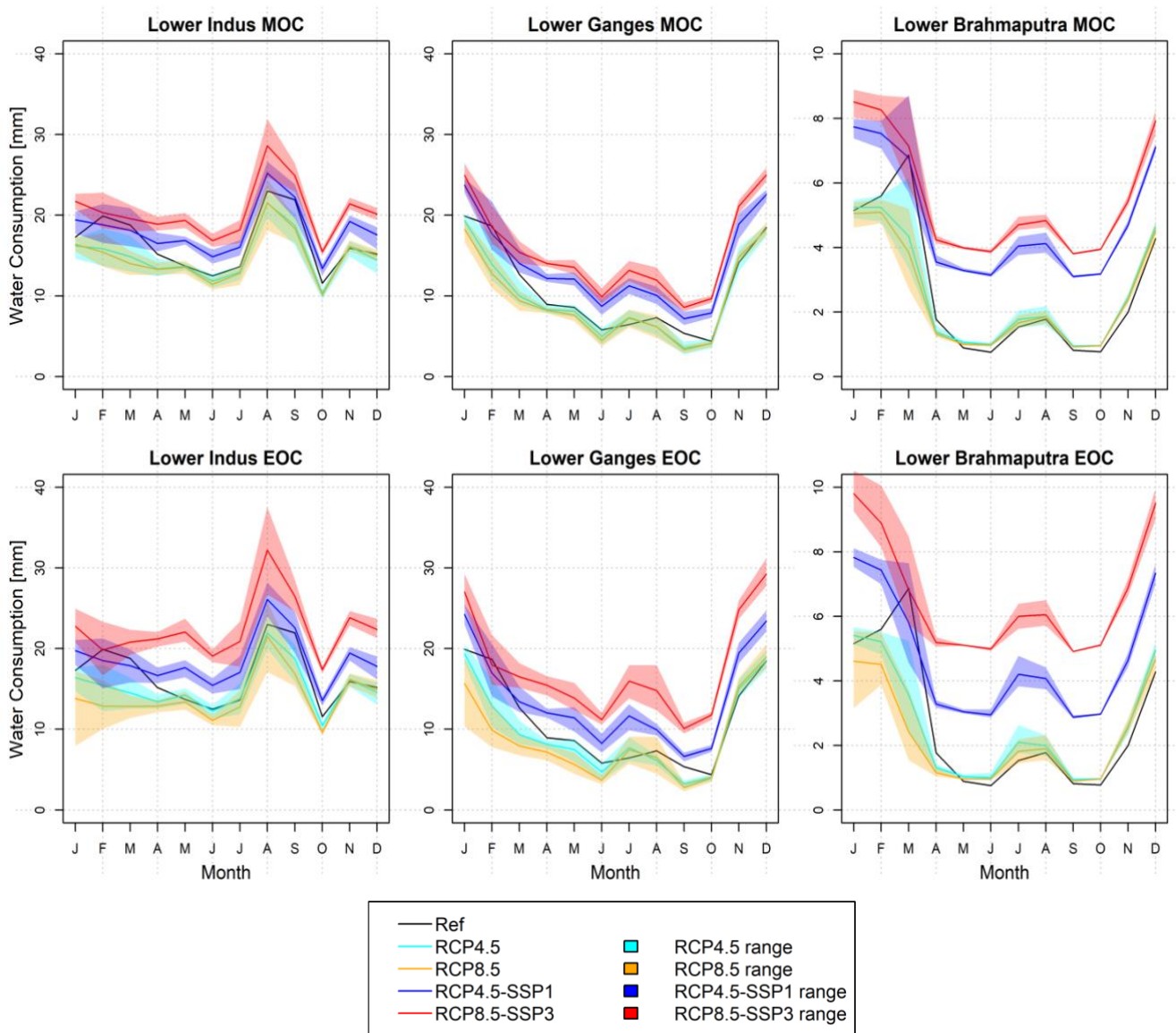

**Figure 7**. Monthly projected changes in the total water consumption for RCP4.5, RCP8.5, RCP4.5 – SSP1, and RCP8.5 – SSP3. The projected changes are given for the mid and end of the 21$^{st}$ century (MOC and EOC). The coloured bands represent the range of ensemble projections that are resulting from forcing the LPJmL model with the different climate models.

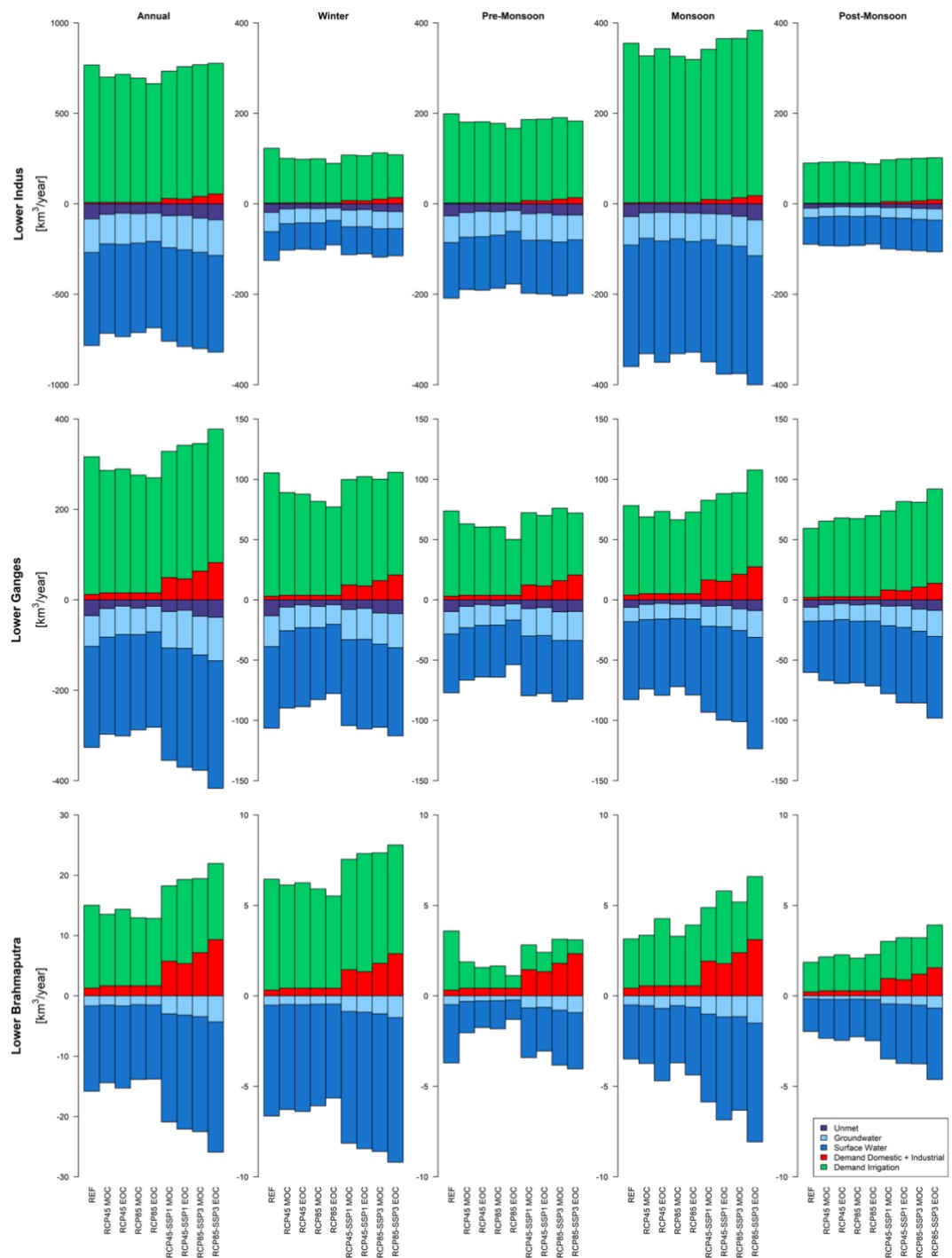

**Figure 8.** Projected changes in the annual and seasonal blue water demand and supply for RCP4.5, RCP8.5, RCP4.5 – SSP1, and RCP8.5 – SSP3. The projected changes are given for the mid and end of the 21$^{st}$ century (MOC and EOC).

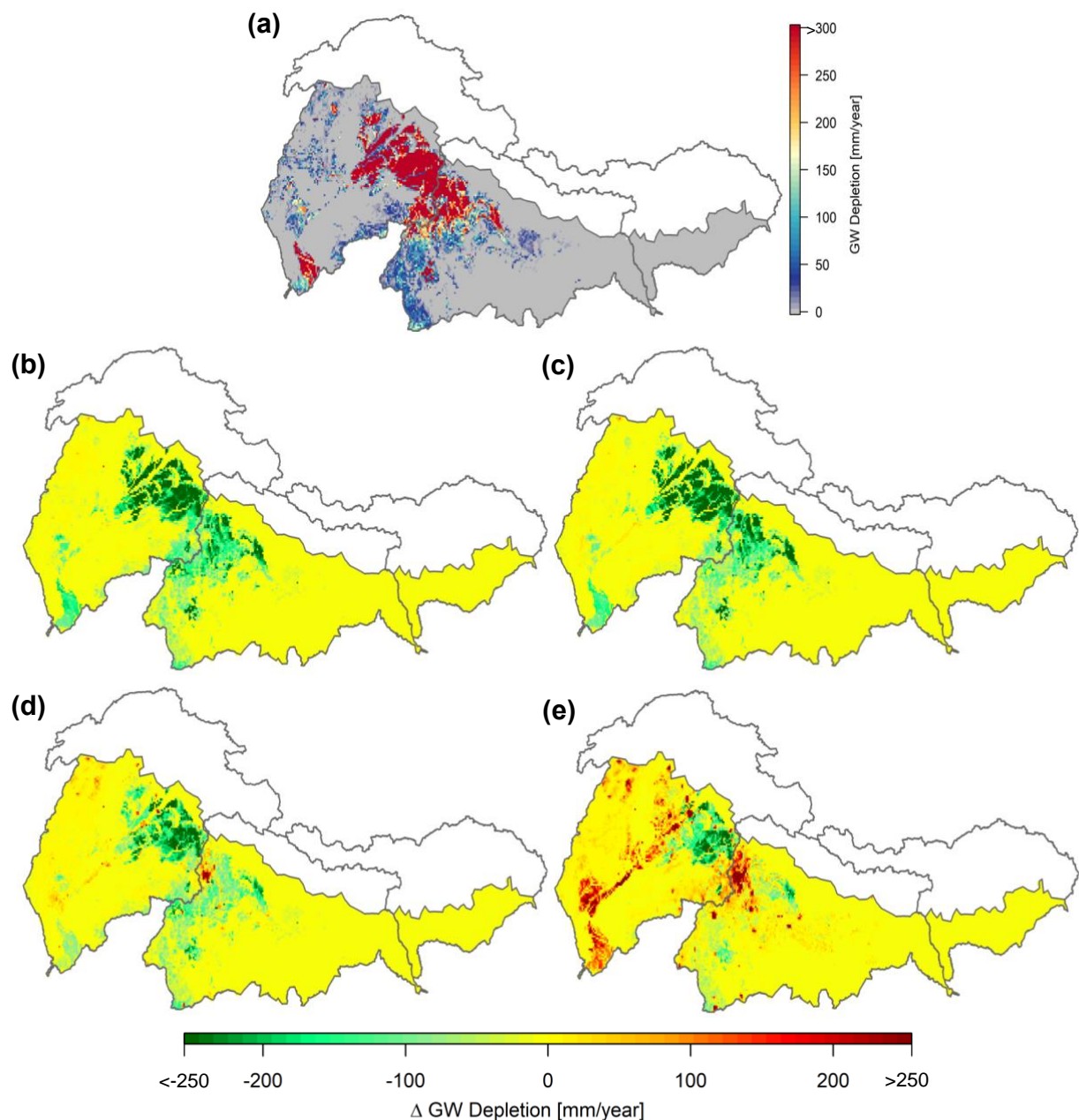

**Figure 9**. Maps showing the annual groundwater depletion for the reference period (a) and the projected changes in groundwater depletion for RCP4.5 (b), RCP8.5 (c), RCP4.5 – SSP1 (d), and RCP8.5 – SSP3 (e). The projected changes are given for the end of the 21st century. Green indicates less depletion and red indicate more depletion.

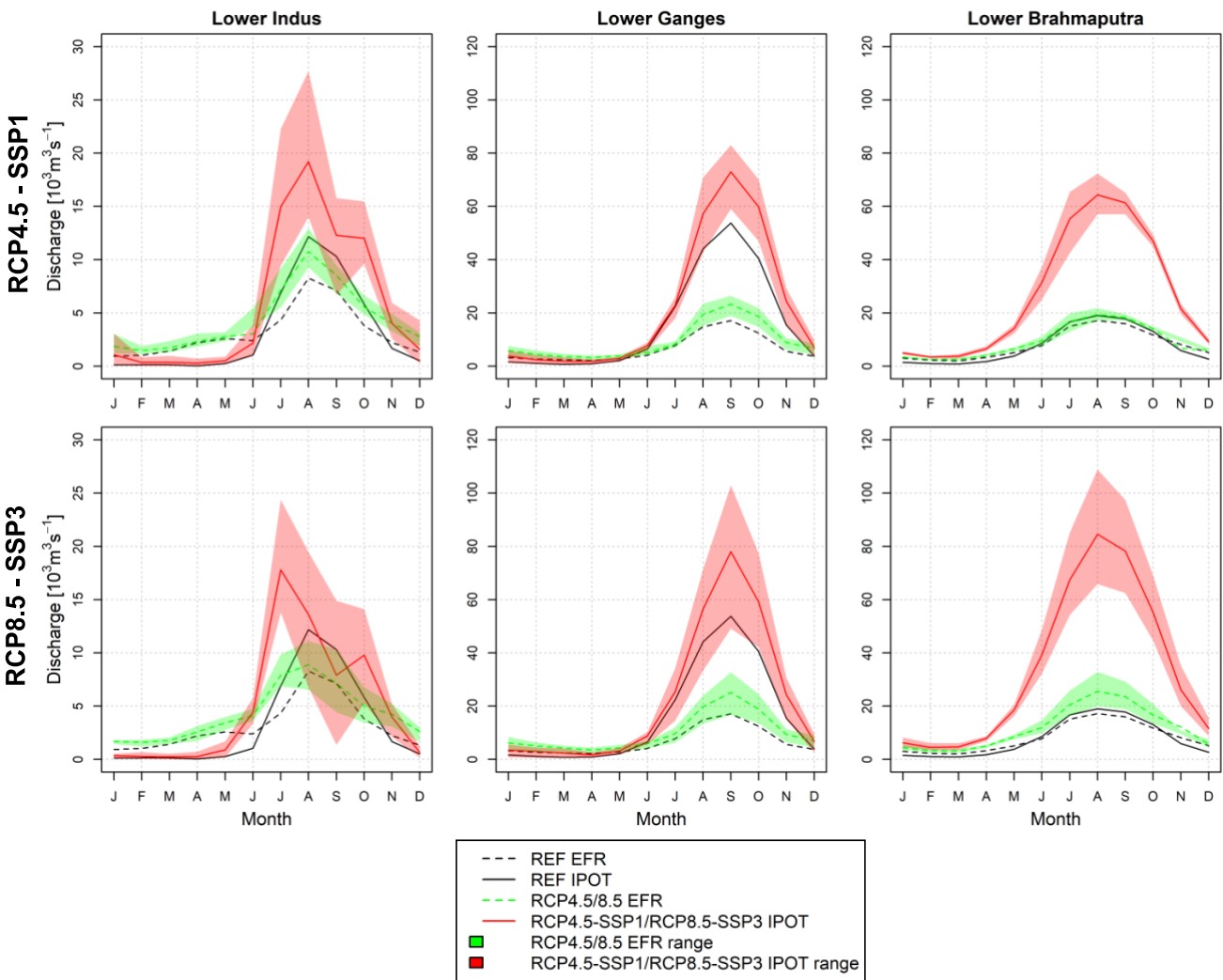

**Figure 10**. Monthly projected changes in the environmental flow requirements (EFR) and anthropogenic influenced discharge (IPOT) at the outlets of the Indus, Ganges and Brahmaputra rivers for RCP4.5 – SSP1 (upper row) and RCP8.5 – SSP3 (lower row). The projected changes are given for the end of the 21$^{st}$ century (EOC). The coloured bands represent the range of ensemble projections that are resulting from forcing the LPJmL model with the different climate models and SSP storylines.

**Table 1.** Projected basin-aggregated population counts and GDP (PPP = Purchasing Power Parity) for SSP1 and SSP3. The population counts are extracted from the HYDE v3.2 database (Klein Goldewijk et al., 2010). The GDP (PPP) is a product of the population counts and the country-specific GDP (PPP) per capita, which is derived from the IIASA SSP database (IIASA, 2017) as the ensemble mean of the IIASA GDP and OECD Environmental Growth models.

| Basins | Countries | Population (x $10^6$) | | | GDP (PPP) (x $10^9$ US$2005) | | |
|---|---|---|---|---|---|---|---|
| | | 2010 | 2050 | 2100 | 2010 | 2050 | 2100 |
| **Indus** | AF, CN, IN, PK | 245 | 346/469 | 289/725 | 631 | 5124/2894 | 14574/7191 |
| **Ganges** | BD, CN, IN, NP | 494 | 629/804 | 466/1073 | 1410 | 14276/8782 | 28796/15198 |
| **Brahmaputra** | BD, BT, CN, IN | 65 | 81/101 | 58/129 | 165 | 1601/952 | 3299/1689 |

**Table 2.** Projected changes in the annual and seasonal blue water gap of the Indus and Ganges river basins under present (1981-2010) and far-future (2071-2100; EOC) conditions for RCP4.5, RCP8.5, RCP4.5-SSP1, and RCP8.5-SSP3. The values between the parentheses represent the standard deviation.

| Basin | Scenario | Annual | Winter | Pre-monsoon | Monsoon | Post-monsoon |
|---|---|---|---|---|---|---|
| Indus | REF [km$^3$] | 83 | 19 | 27 | 28 | 10 |
| | RCP45 EOC [%] | -36 (15) | -47 (14) | -35 (16) | -32 (15) | -33 (15) |
| | RCP85 EOC [%] | -37 (15) | -46 (11) | -37 (11) | -34 (21) | -30 (18) |
| | RCP45-SSP1 EOC [%] | -21 (18) | -31 (18) | -21 (20) | -16 (18) | -15 (18) |
| | RCP85-SSP3 EOC [%] | 7 (25) | -11 (18) | -9 (8) | 30 (52) | 18 (24) |
| Ganges | REF [km$^3$] | 35 | 13 | 10 | 6 | 6 |
| | RCP45 EOC [%] | -52 (21) | -61 (17) | -51 (23) | -44 (23) | -41 (27) |
| | RCP85 EOC [%] | -55 (20) | -64 (16) | -54 (19) | -47 (28) | -44 (25) |
| | RCP45-SSP1 EOC [%] | -23 (32) | -37 (27) | -23 (33) | -9 (35) | -8 (39) |
| | RCP85-SSP3 EOC [%] | 14 (43) | -11 (36) | 1 (26) | 55 (75) | 50 (55) |