# Peer review of "Climate change vs. socio-economic development: Understanding the future South Asian water gap"

_Hydrology and Earth System Sciences, 2018_

## Editor Comment (EC1) · P. P. Mujumdar (Editor) · 27 Mar 2018

Dear Authors,

We notice that Figure 5 is missing in your manuscript file. Pl. upload the manuscript again, complete in all respects. Pl. keep the editorial office informed, if you are uploading the file afresh.

Best wishes,

Pradeep Mujumdar (Handling Editor)

---

## Referee Comment (RC1) · Anonymous Referee #1 · 8 May 2018

Comments on: "Climate change vs. socio-economic development: Understanding the future South-Asian water gap" Authors: Wijngaard et al. Submitted to: HESS Article ref no.: hess-2018-16

The research in the article under review is an attempt to predict the combined impact of climate change and socio-economic changes on water scarcity in the downstream portions of 3 rivers originating in the Himalayas: the Indus, the Ganga and the Brahmaputra. It seeks to improve upon earlier predictive modelling by using a larger ensemble of climate predictions, separate hydrological models for upstream (hill) and downstream (plains/delta) regions, and more careful simulation of agricultural water

use in the downstream region through using recently developed models that simulate distribution through canal systems and timing of water demand in multiple cropping systems. It also draws upon recently published Shared Socioeconomic Pathways (SSP) developed by the climate change community that describe alternative socio-economic development scenarios.

The technical side of the research has been done quite competently for the most part and the writing is also mostly clear and well organized. My concerns with the paper mostly are at the macro-level, viz., as to what (value) assumptions it makes in framing the research, and what contribution it makes to our understanding of the likely outcomes of multiple-stressors operating in the study region. Also a couple of concerns about the modelling.

1. Water uses considered: The authors only take into consideration water use in agriculture, industry and the domestic sectors. In doing so, they leave out in-stream environmental (and fishing) needs, as well as minimum ecosystem flows that need to go out to the ocean. This framing creates the impression that it is 'okay' to consume all the surface flow, which is problematic. Given the higher temporal and spatial resolution that the models have incorporated, the authors can easily provide for these other uses also. Which uses to cater to is of course a value-loaded decision, but no more than the decisions already made. The authors could allow for variation in societal values by showing the tradeoffs between (e.g.) meeting minimum ecosystem flow standards (that might affect agricultural production) and prioritising agricultural needs (thereby violating minimum flows).

2. Definition of water scarcity: The manner in which water scarcity is defined (with respect to the above 3 uses) is in terms of a 'blue water gap' (gap between supply and demand of blue water) which then manifests itself as over-extraction of groundwater. But over-extraction has inter-temporal effects, so it seems that this is a definition of 'unsustainability'. On the other hand, the manner in which groundwater overextraction manifests itself is in the form of loss of base flows, which means either loss to agri/domestic users downstream or loss to instream/ocean uses. Neither of which is captured here. Scarcity can be the outcome of distributional issues unrelated to absolute availability. So one definition of scarcity could have been the fraction of the population (in each sector) facing water shortages. More generally, 'scarcity' is a social construct, and if the research is to be useful to policy-makers in the region, the 'outcome variable' in the modelling must reflect local, multiple understandings of scarcity.

3. Contribution: The need to model the impact of multiple stressors rather than of climate change in isolation has now been recognized in the water resources community for a while. In a well-known coarse-scale analysis, Vorosmarty et al.(2000) pointed out that rising human demand for water will outweigh the impacts of climate change on water resources in the south Asian region. [The authors appear to have misinterpreted this study in p.2 line 30: Vorosmarty et al conclude "that impending global-scale changes in population and economic development over the next 25 years will dictate the future relation between water supply and demand to a much greater degree than will changes in mean climate."] . So the question is in what way does this study deepen our understanding of this broad trend or likely responses?

My assessment is "not much, given the uncertainties involved and the limitations of the approach used" [uncertainties are also discussed below]. That climate change is predicted to increase water availability in all 3 basins is clear, once one reads the CC predictions for this region from the GCM runs chosen. That socio-economic developments will (in the absence of any adaptive responses) lead to increases in water demand is obvious to anyone who knows the region. The net result is that "The combination of climate change and socio-economic development is expected to result in increasing water gaps with relative increases up to 7% and 11% in the Indus and Ganges, respectively" [p.18, line 27]". To my mind, this small change is well within the errors/uncertainties of all the modelling that has been done. (Since these results are not presented in tabular form but only in the bar charts in Figure 8, it is even hard to see that the water gap has actually increased vis-à-vis the reference scenario.) So one is unable to see the value of such a coarse result. The lack of endogeneity in the modelling framework (i.e., the fixed nature of landuse predictions driven by population growth and economic change and the lack of any adaptive response by any water user to water scarcity) means that we are unable to see to what extent adaptive actions might ameliorate the problem. And the lack of information on "who actually suffers because of the scarcity" prevents the analysis from throwing up any interesting social impact information.

4. Modelling: There are some concerns with the manner in which the modelling has been done. They may not all affect the results seriously, but need to be tabled and discussed:

a. Groundwater is treated as being separate from surface water. E.g., page 2, line 3 says the 3 sources of water are rainfall-runoff, groundwater and meltwater. This would be true if runoff did not include baseflows, which is groundwater returning to the surface as discharge (see Ponce, V.M., 2007. Sustainable yield of groundwater. http://ponce.sdsu.edu/groundwater_sustainable_yield.html, and Sophocleous, M., 2000. From safe yield to sustainable development of water resources—the Kansas experience. Journal of Hydrology 235, 27–43.). But then on page 7, line 2, the authors say total runoff is sum of glacier & snow runoff (presumably melt), surface runoff, lateral flow and baseflow. This then leads to double counting, since "water for irrigation and other uses can be drawn from surface water in the grid cell [which would include baseflows], etc etc. and groundwater bodies" (page 8, line 10-12).

b. This treatment of GW as separate from SW also enables the authors to talke of the blue water gap in terms of unsustainable withdrawal of GW (i.e., withdrawal more than recharge) without realizing that the first impact of such over-withdrawal is the loss of baseflows, which will affect downstream grid cells. GW depletion is not a separate/separable phenomenon, unless one is talking about depleting non-renewable forms of GW.

c. The 'daily timestep' is clearly a case of spurious precision. Water use is definitely

not known/predictable at such a fine temporal scale.

d. The assumption that 'water availability in upstream regions' (the Himalayan catchments) is dependent upon natural factors' may be true for the Indus and the Brahmaputra, but questionable for the Ganga basin. Uttarakhand and Nepal are witnessing massive interventions in hydrology in the form of dams (large and small) as well as traditional uses for agriculture—these regions a dense network of community-scale irrigation systems.

e. To the best of my knowledge, the SPHY model has very litte stream gauge/river gauge data available (at least in the Ganga basin) to validate itself. So there must be major uncertainties just with the flow predictions for the 'upstream' model.

f. The predictions under climate change and SSP are compared with the 'reference' period results, which seem to be the average of the period 1981-2010. This is a lengthy period over which major changes have taken place in the water resource use in this region, and using an average for this whole period makes it unusable as a 'reference'.

Minor technical and editorial comments are given in the marked up pdf attached herewith.

Please also note the supplement to this comment:
https://www.hydrol-earth-syst-sci-discuss.net/hess-2018-16/hess-2018-16-RC1-supplement.pdf
* * *
[Figure]

**Supplement:**

[revised manuscript text omitted]

---

## Editor Comment (EC2) · P. P. Mujumdar (Editor) · 26 May 2018

I agree with the major comments and concerns of the anonymous reviewer. In addition, I would like the authors to add a brief discussion in the "Uncertainties and Limitations" section on the possible non-stationarity in the hydrologic model parameters because of the change in climate and landuse.
* * *

---

## Author Comment (AC2) · 9 Jul 2018

We greatfully acknowledge the reviewer for his/her remarks and suggestions, which improved the quality of the manuscript significantly. We have carefully considered the suggestions of the reviewer and we provide a point-by-point response to the reviewer's comments. For clarity, the reviewer's comments are given in italics and the responses are given in plain text. References that do not refer to those in the main manuscript are listed below. The manuscript will be modified accordingly to the responses that are given to the comments.

*The research in the article under review is an attempt to predict the combined impact of climate change and socio-economic changes on water scarcity in the downstream portions of 3 rivers originating in the Himalayas: the Indus, the Ganga and the Brahmaputra. It seeks to improve upon earlier predictive modelling by using a larger ensemble of climate predictions, separate hydrological models for upstream (hill) and downstream (plains/delta) regions, and more careful simulation of agricultural water use in the downstream region through using recently developed models that simulate distribution through canal systems and timing of water demand in multiple cropping systems. It also draws upon recently published Shared Socioeconomic Pathways (SSP) developed by the climate change community that describe alternative socio-economic development scenarios.*

*The technical side of the research has been done quite competently for the most part and the writing is also mostly clear and well organized. My concerns with the paper mostly are at the macro-level, viz., as to what (value) assumptions it makes in framing the research, and what contribution it makes to our understanding of the likely outcomes of multiple-stressors operating in the study region. Also a couple of concerns about the modelling.*

Thank you. We have tried to address all concerns and our detailed response is provided below.

1. *Water uses considered: The authors only take into consideration water use in agriculture, industry and the domestic sectors. In doing so, they leave out in-stream environmental (and fishing) needs, as well as minimum ecosystem flows that need to go out to the ocean. This framing creates the impression that it is 'okay' to consume all the surface flow, which is problematic. Given the higher temporal and spatial resolution that the models have incorporated, the authors can easily provide for these other uses also. Which uses to cater to is of course a value-loaded decision, but no more than the decisions already made. The authors could allow for variation in societal values by showing the trade-offs between (e.g.) meeting minimum ecosystem flow standards (that might affect agricultural production) and prioritising agricultural needs (thereby violating minimum flows).*

We indeed only consider water use in the agricultural, domestic, and industrial sectors. We consider water use in these sectors, because these sectors are the largest water consumers in South Asia (FAO, 2016). Although there is a version of LPJmL that does allocate water requirements of aquatic ecosystems and its trade-off with food production (Jägermeyr et al., 2017), we decided not to impose any restrictions to withdrawals in this study, because we believe that the water requirements by ecosystems are not considered in this region (yet). Since the Indus, Ganges, and Brahmaputra river basins are facing rapid and continuous population growth, food demand will increase and a higher agricultural production is required. Therefore, agricultural needs will probably be prioritized at the cost of environmental flows. To investigate which impact (future) blue water consumption has on environmental flow transgressions we have however added an extra subsection "Environmental Flows" to the Results section. The outcomes in this subsection show that the transgression of future environmental flows will likely be limited with sustained environmental flows requirements (EFRs) during the monsoon season and unmet EFRs during the low flow season in the Indus and Ganges river basins. Further we discuss environmental flows and the use of EFRs in the Introduction section and will add an extra subsection "Analysis of Environmental Flows" to the Methods section to describe the methodology behind the derivation of EFRs.

2.  *Definition of water scarcity: The manner in which water scarcity is defined (with respect to the above 3 uses) is in terms of a 'blue water gap' (gap between supply and demand of blue water) which then manifests itself as over-extraction of groundwater. But over-extraction has inter-temporal effects, so it seems that this is a definition of 'unsustainability'. On the other hand, the manner in which groundwater overextraction manifests itself is in the form of loss of base flows, which means either loss to agri/domestic users downstream or loss to instream/ocean uses. Neither of which is captured here. Scarcity can be the outcome of distributional issues unrelated to absolute availability. So one definition of scarcity could have been the fraction of the population (in each sector) facing water shortages. More generally, 'scarcity' is a social construct, and if the research is to be useful to policy-makers in the region, the 'outcome variable' in the modelling must reflect local, multiple understandings of scarcity.*

The influence of groundwater (over-)abstraction on the amount of baseflow and subsequent availability for downstream agricultural, domestic, and industrial users are captured in the model. In the LPJmL model groundwater reservoirs are replenished by seepage from the bottom soil layers and contribute to the river network by means of baseflow. Baseflow is calculated by means of a linear reservoir function, where groundwater only adds to baseflow when groundwater recharge is larger than groundwater withdrawal. When groundwater recharge is smaller than groundwater withdrawal, no baseflow does occur, which results in a loss of streamflow. This will eventually have an impact on downstream water availability. We will add an extra sentence to the model description of LPJmL to clarify the relation between groundwater bodies and the river network.

We agree with the reviewer that water scarcity can indeed be understood as the outcome of distributional issues unrelated to absolute water availability and therefore can be defined as the lack of access to adequate quantities of water that are needed to fulfil water requirements. The blue water gap as defined in our study is a way to measure the water scarcity by means of unsustainable groundwater withdrawal, which in the LPJmL model can only occur when surface water or renewable groundwater is not available. That surface water or renewable groundwater is not available locally makes that there are no adequate quantities of water and is thus an indication for water scarcity. The main advantage of using the blue water gap as indicator is that we can account for all spatial and temporal mismatches between water demand and supply, which is therefore a way to show the heterogeneity in water scarcity throughout the basins. The magnitude of the water gap is highly depending on the presence of local reservoirs and canal systems, and on the area where the water gap occurs. Figure 9 illustrates very well that the water gap is especially an issue in urban areas, whereas in rural areas the water gap is limited. In combination with the identification on which drivers and processes are responsible for the development of the water gap, this can provide valuable information for policy makers, and shows that the water gap similar to water scarcity can be interpreted as a social construct. We agree with the reviewer that the fraction of the population facing water shortages could have been a way to measure water scarcity. However, according to Liu et al. (2017) there are also other indicators that can be used to measure water scarcity, which can lead to different outcomes. The large variety in indicators makes it therefore difficult to compare the outcomes of one study with the other. To have a robust representation of water scarcity future studies might be needed that assess water scarcity by using a set of multiple indicators. We will add this point to the discussion. Further we will change the manuscript to put more emphasis on assessing the future evolution of the blue water gap and not the future blue water scarcity, which is a broader term.

3.  *Contribution: The need to model the impact of multiple stressors rather than of climate change in isolation has now been recognized in the water resources community for a while. In a well-known coarse-scale analysis, Vorosmarty et al.(2000) pointed out that rising human demand for water will outweigh the impacts of climate change on water resources in the south Asian region. [The authors appear to have misinterpreted this study in p.2 line 30: Vorosmarty et al conclude "that impending global-scale changes in population and economic development over the next 25 years will dictate the future relation between water supply and demand to a much greater degree than will changes in mean climate."]. So the question is in what way does this study deepen our understanding of this broad trend or likely responses?*

*My assessment is "not much, given the uncertainties involved and the limitations of the approach used" [uncertainties are also discussed below]. That climate change is predicted to increase water availability in all 3 basins is clear, once one reads the CC predictions for this region from the GCM runs chosen. That socio-economic developments will (in the absence of any adaptive responses) lead to increases in water demand is obvious to anyone who knows the region. The net result is that "The combination of climate change and socio-economic development is expected to result in increasing water gaps with relative increases up to 7% and 11% in the Indus and Ganges, respectively" [p.18, line 27]". To my mind, this small change is well within the errors/uncertainties of all the modelling that has been done. (Since these results are not presented in tabular form but only in the bar charts in Figure 8, it is even hard to see that the water gap has actually increased vis-à-vis the reference scenario.) So one is unable to see the value of such a coarse result. The lack of endogeneity in the modelling framework (i.e., the fixed nature of landuse predictions driven by population growth and economic change and the lack of any adaptive response by any water user to water scarcity) means that we are unable to see to what extent adaptive actions might ameliorate the problem. And the lack of information on "who actually suffers because of the scarcity" prevents the analysis from throwing up any interesting social impact information.*

We agree with the reviewer that we misinterpreted the outcomes of Vörösmarty et al. (2000). We have corrected this accordingly.  We also agree with the reviewer that the need to model the impact of multiple drivers rather than climate change in isolation has been recognized in the water resources community. There have been several global studies, including those from Vörösmarty et al. (2000) and Hanasaki et al. (2013), that have investigated the impacts of climate change and/or socio-economic development on water scarcity. These studies have concluded that rising water demands as a response to socio-economic developments are a more important driver than climate change. This might also be obvious for the Indus, Ganges, and Brahmaputra river basins. However, there has not been a study so far that conducted a high resolution integrated assessment on the evolution of the future blue water gap in the Indus, Ganges, and Brahmaputra river basins, thereby aiming at quantifying the impacts of climate change and/or socio-economic development on the regional water gap. The main difference with global studies, such as the study of Vörösmarty et al. (2000), is that a) a coupled modelling approach is applied for the entire IGB that includes a high-resolution cryospheric-hydrological model that can simulate upstream water availability (and represents mountain-hydrological processes that are important in the region), and a high-resolution hydrology and crop production model that can simulate the downstream water availability, supply, demand, and gap; b) the hydrology and crop production model that is applied for downstream domains has specially been developed for this region by including human interventions, such as the extensive irrigation canal systems of the Indus and Ganges river basins, and  multiple cropping systems; c) the high resolution models are forced with an ensemble of downscaled and bias-corrected GCMs, that represents a wide range of possible futures in terms of regional climate change for RCP4.5 and RCP8.5, in combination with SSP storylines. These novelties are highlighted in the last paragraph of the Introduction.

We made one miscalculation. The ensemble mean of the projected changes in the water gap of the Ganges river basin is 14% instead of 11%. The relative changes of 14% and 7% that are projected for the Ganges and Indus river basins, respectively are indeed within the uncertainty range of model outcomes that are generated for the different climate models in combination with SSP storylines. We agree with the reviewer that it is difficult to read the changes in the water gap from Figure 8. We will include an extra table that lists the ensemble means and standard deviations of the projected changes in the future blue water gap at the end of the 21st century.

We are aware that the lacking endogeneity of land use and adaptive responses of water users in the modelling framework are likely to introduce uncertainties in the outcomes of the model. This study aims however at providing a first comprehensive integrated assessment that identifies the main processes and drivers that are responsible for changes in the future water gap. Investigating the impact of adaptation strategies on the future water gap is beyond the scope of this study and needs further investigation in the future by assessing the potentials of well-informed realistic adaptation strategies (i.e. that have been developed by means of piloting) in closing the water gap. We will add a paragraph to the discussion on the uncertainties that can emerge due to the lacking adaptive response. The uncertainties due to lacking endogeneity are mentioned in the discussion on uncertainties and limitations.

4. *Modelling: There are some concerns with the manner in which the modelling has been done. They may not all affect the results seriously, but need to be tabled and discussed:*

   a) *Groundwater is treated as being separate from surface water. E.g., page 2, line 3 says the 3 sources of water are rainfall-runoff, groundwater and meltwater. This would be true if runoff did not include baseflows, which is groundwater returning to the surface as discharge (see Ponce, V.M., 2007. Sustainable yield of groundwater. http://ponce.sdsu.edu/groundwater_sustainable_yield.html, and Sophocleous, M., 2000. experience. Journal of Hydrology 235, 27–43.). But then on page 7, line 2, the authors say total runoff is sum of glacier & snow runoff (presumably melt), surface runoff, lateral flow and baseflow. This then leads to double counting, since "water for irrigation and other uses can be drawn from surface water in the grid cell [which would include baseflows], etc etc. and groundwater bodies" (page 8, line 10-12).*

We categorize groundwater and surface water according to definitions that are consistent with commonly used definitions of the FAO and other water use studies, such as Siebert et al. (2010). Thereby, groundwater is defined as water that is abstracted from groundwater bodies (i.e. shallow/deep aquifers) by using (artificial) wells, and surface water as water that is abstracted directly from rivers, lakes, and reservoirs. Groundwater that has not been withdrawn locally will add to baseflow and enters surface water by means of drainage. From the moment baseflow enters surface water it is considered as surface water when it is used for downstream water supply. We listed these definitions in Section 3.1 Definitions, but will update the definitions to clarify what we define as groundwater and as surface water.

On page 2, line 3 we indeed mention that the water supply has 3 main components: (monsoon) rainfall-runoff, groundwater, and meltwater. The (monsoon) rainfall-runoff as mentioned in this sentence does not include baseflow, but is defined as the surface component of the total runoff that result from rainfall. We are however aware that this sentence may lead to confusion. Therefore, we will change this sentence and mention that water supply is dominated by two different components: locally pumped groundwater and surface water supplied by irrigation canals. Thereby surface water supplied by irrigation canals that are diverted from rivers and reservoirs has three main constituents: direct rainfall runoff, meltwater from upstream located ice and snow reserves, and baseflow.

The total runoff as described on page 7, line 2 is calculated by the SPHY model, which only simulates upstream water supply and does not include the effects of agriculture/irrigation on water supply. In SPHY, surface runoff is not the same as the baseflow runoff, and are both separate components of the total runoff, which means in SPHY baseflow can be a component of surface water. In downstream domains, we apply the LPJmL model (as described on page 7 line 21-33, and page 8 line 1-14). In LPJmL, the contribution from groundwater to surface water is simulated by means of drainage, which means baseflow is a component of surface water as well. As long groundwater is withdrawn from a grid cell it is not added to the baseflow, which means upstream groundwater withdrawals can affect downstream water availability by decreasing the baseflow contribution. Because of this groundwater and surface water can be treated separately, and double counting does not occur.

   b) *This treatment of GW as separate from SW also enables the authors to talke of the blue water gap in terms of unsustainable withdrawal of GW (i.e., withdrawal more than recharge) without realizing that the first impact of such over-withdrawal is the loss of baseflows, which will affect downstream grid cells. GW depletion is not a separate/separable phenomenon, unless one is talking about depleting non-renewable forms of GW.*

In our model, the over-abstraction of groundwater does influence the amount of baseflow and thus the amount of water available for downstream grid cells. As mentioned under comment 4a, groundwater that is withdrawn cannot be added to the baseflow, which means groundwater withdrawals result in a baseflow reduction and subsequently will affect downstream water availability.

*c) The 'daily timestep' is clearly a case of spurious precision. Water use is definitely not known/predictable at such a fine temporal scale.*

We agree with the reviewer that it is difficult to predict water scarcity on daily basis. However, to predict water scarcity accurately in river basins that highly depend on upstream (mountain) water resources it is important to have a robust representation of mountain-hydrological processes that are highly variable in space and time. In particular, glacier and snow melt processes are not sufficiently captured at larger time steps. For this reason, it is needed to use hydrological models on a high temporal resolution (i.e. daily). The outcomes however, are aggregated to monthly time steps to assess the monthly and seasonal variations in the water availability, supply, demand, and gap.

*d) The assumption that 'water availability in upstream regions' (the Himalayan catchments) is dependent upon natural factors' may be true for the Indus and the Brahmaputra, but questionable for the Ganga basin. Uttarakhand and Nepal are witnessing massive interventions in hydrology in the form of dams (large and small) as well as traditional uses for agriculture these regions in a dense network of community-scale irrigation systems.*

We are aware that in the upstream domains also human interventions (such as dams, reservoirs, and irrigation) take place and that this also can influence the hydrological cycle, which means water availability is not fully dependent on 'natural factors'. However, we assume the (current) impact of dams/reservoirs and agriculture to be low in comparison with the influence dams/reservoirs and agriculture have in the downstream domains. For instance, compared to the downstream domains the number of dams/reservoirs is limited. The province the reviewer mentioned as example (i.e. Uttaranchal/Uttarakhand) is among the upstream areas the area with the most dams and capacity (i.e. 15 dams with a total capacity of about 5 $km^3$) (FAO, 2016). In other upstream areas, the number of dams is rather small and the total capacity is low compared to the larger dams in downstream domains. For instance, Tarbela dam has a total capacity of 12 $km^3$, whereas the total capacity of dams in the upstream domains reach up to about 5.5 $km^3$ distributed over about 50 dams (FAO, 2016). Furthermore, most dams are designed as hydropower dams with limited storage or as run-off-the-river hydropower plants, which have a low degree of regulation in the upstream domains of the IGB (Lehner et al., 2011; FAO, 2016). Further it has been found in earlier studies (Biemans et al., 2016) that the irrigation water demand (and cropping intensity) in upstream areas is rather low (i.e. <100 mm $yr^{-1}$) in comparison with the irrigation water demand (and cropping intensity) in downstream areas (i.e. >500 mm $yr^{-1}$). We agree with the reviewer that the absence of human interventions in the SPHY model introduces uncertainties in the amount of water that is available for downstream areas. Therefore, we will add a point to the discussion on to point out this drawback. Further we will also point out that future planned infrastructure need to be included in future work to assess the impact on the blue water gap.

*e) To the best of my knowledge, the SPHY model has very litte stream gauge/river gauge data available (at least in the Ganga basin) to validate itself. So there must be major uncertainties just with the flow predictions for the 'upstream' model.*

We have calibrated and validated the model by means of a three-step systematic approach on upstream and downstream located gauging stations in the three consecutive upstream domains of the IGB (Wijngaard et al., 2017). This approach comprised of the following steps: 1) a calibration on geodetically derived glacier mass balances, 2) MODIS-derived snow cover maps, and 3) a calibration on observed discharge of six gauging stations, namely Dainyor Bridge (upstream UIB), Besham Quila (downstream UIB), Bimalnagar (upstream UGB), Devghat (downstream UGB), Wangdirapids (upstream UBB), and Sunkosh (downstream UBB). This three-step calibration approach was implemented to reduce calibration problems of equifinality (Pellicciotti et al., 2012). The calibration and validation of SPHY resulted in model performances with Nash Sutcliffe Efficiency values between 0.60 ("satisfactory") for Dainyor Bridge and 0.84 ("very good") for Devghat. We agree with the reviewer that there can be uncertainties in the streamflow for some of the outflow points of the upstream domains due to different

basin characteristics. For a more detailed discussion on uncertainties that are potentially included in the streamflow data, we refer to Wijngaard et al. (2017).

> *f) The predictions under climate change and SSP are compared with the 'reference' period results, which seem to be the average of the period 1981-2010. This is a lengthy period over which major changes have taken place in the water resource use in this region, and using an average for this whole period makes it unusable as a 'reference'.*

We are aware that a 30-year period is a lengthy period over which major changes can occur in water use in this region. We have decided to use a 30-year period, because this is common and recommended when investigating climate change impacts. Also in other high impact studies (e.g. Hanasaki et al., 2013; Wada et al., 2013) that investigated current and future water scarcity a period of 30 years is used in the analysis of outcomes, also for the calculation of single averages.

*Minor technical and editorial comments are given in the marked up pdf attached herewith.*

Minor and Technical Comments

*1. P2 L1: What about conservation purposes? Cultural purposes?*
According to the FAO AQUASTAT Database (FAO, 2016), the agricultural, domestic, and industrial sectors are the largest water consumers. Cultural water use is related to the amount of water used per capita in each country specifically, and can therefore be interpreted to be part of domestic water use. Requirements for conservation activities have a very low rate of water consumption (FAO, 2016) and have therefore a very low share in the total water consumption in comparison with the agricultural, domestic, and industrial water sectors. To mention that these sectors are the dominant water users, but are not the only water users, we have adapted the sentence to clarify that water is *mainly* used for agricultural, domestic, and industrial purposes.

*2. P2 L3: Not separate from surface water. Unless one is talking about non-renewable sources...*
We refer to comment 4a.

*3. P2 L18: CC will also affect GW when rainfall changes*
We agree with the reviewer that climate change will also affect groundwater when rainfall changes. We will add an extra sentence to the manuscript to clarify that long-term precipitation changes may lead to changes in groundwater recharge and storage, and thus may affect groundwater availability. These processes are all included in our modelling system.

*4. P2 L23: Mentioning only one possible driver (demographic pressure) makes the approach sound very Malthusian. Economic growth/industrialisation, urbanization, shifting agricultural patterns and trade, and governance issues (including inter-nation conflict) can be as big or bigger drivers as population. These in turn result in human interventions on a variety of scales, including large dam projects and interlinking of rivers-type mega-projects. Since later on in the paper you have taken a broader approach in talking about the multiple stressors, I suggest doing so consistently from the beginning.*
We agree with the reviewer that multiple drivers need to be mentioned in the Introduction since the paper also focusses on multiple drivers. Therefore, we will add a few extra sentences to the Introduction that also points out the future changes that are related to the other drivers: economic growth, industrialization, urbanization, and intensification of water use in food production resulting from changes in agriculture.

*5. P2 L34: May depend upon how scarcity is defined and measured, rather than difference in prediction quality or rigour.*
We refer to comment 2

*6. P3 L9: What about all the other drivers mentioned above? (And some not mentioned?) The real source of uncertainty is because of uncertainty in development pathways and outcomes*

This part of the introduction points out a few examples of the drawbacks of the approaches used in the other studies. We agree with the reviewer that drawbacks related to other drivers also should be mentioned. Therefore, we will include this point into the Introduction to mention drawbacks related to socio-economic drivers. Further, the study of Arnell & Lloyd-Hughes (2014), who investigated the relative contribution of SSPs, RCPs, and climate models to the regional and global impact on absolute exposure to increased water resources scarcity, found that in South Asia the uncertainty related to climate models has the largest relative contribution in comparison with the uncertainty related to RCPs and SSPs. To cover the uncertainties related to the climate models, RCPs and SSPs, we have selected climate models that represent a wide range of possible futures in terms of climate change, and we have included two contrasting RCPs and SSPs.

*7. P3 L15: How can daily scarcity be predicted if even utilities don't know what it is on a daily basis? infrastructure mediates between daily fluctuations in runoff or rainfall and creates delayed and ameliorated effects.*

We refer to comment 4c

*8. P3 L22 & 26: Huge amount of human intervention now happening in upstream (such as Uttarakhand) makes this distinction less tenable? Uttarakhand alone has 15 dams in Gangetic basin, and many more planned/under construction. In addition, there is substantial amount of agriculture in the upstream domain, the impact on which also needs to be understood.*

We refer to comment 4d.

*9. P4 L1: Projections from whom?*

The projections are from the study of Gain and Wada (2014). We have added a reference to this sentence.

*10. P5 L5: Valid point. But a) daily timestep is unrealistic (false precision), and b) only high temporal resolution without comparable spatial resolution is not very useful*

For comment a) we refer to comment 4c. We decided to set up a model on 5 x 5 arc-min, which is a decision based on the spatial resolution of the available datasets. The climate forcings have a spatial resolution of 10 x 10 km, the water demand fields have a spatial resolution of 5 x 5 arc-min. We are aware that with a higher resolution model more accurate outcomes can be achieved. The disadvantage of a higher resolution model is however that model calculation times become larger, which is therefore considered to be less feasible. In addition, higher resolution models would create a false precision since some of the datasets (e.g. the MIRCA2000 dataset) are only available at a 5 x 5 arc-min spatial resolution. Further, the spatial resolution that has been used for our model set-up is already high compared to the spatial resolution that has been used in other (global) studies, which often use a spatial resolution of 0.5° x 0.5° (i.e. ~50 x 50 km) for their model set-ups (e.g. Vörösmarty et al., 2000; Hanasaki et al., 2013; Gain and Wada, 2014).

*11. P4 L16: Needs clarification*

We have changed the sentence for clarification.

*12. P4 L21: This is 1.8km x 5 = 9km N-S and variable width: is this adequately 'high-res' for such a densely populated region with high spatial variability?*

We refer to minor comment 10.

*13. P4 L22: WHY MODEL ENTIRE IGB together? What is the connectivity between I & GB? Or even between G & B?*

We investigate future water scarcity for the Indus, Ganges, and Brahmaputra river basins, because these river basins are the three major river basins of South Asia that have a direct connectivity with the Hindu-Kush Himalayan mountain range and therefore are interesting to study because of their dependency on mountain water resources. The three basins are investigated because of the contrasting differences in basin characteristics (hydro-climatic and socio-economic) and the way how basins respond to future

climate change and socio-economic developments. The Indus basin for instance a very dry arid downstream climate and a large dependency on upstream water resources. The climate is dominated by westerly disturbances and monsoon systems, bringing precipitation in winter and summer. The climate of Ganges and Brahmaputra basins are dominated by the monsoon bringing large amounts of precipitation during summer. The dependency on upstream water resources is large in the Ganges basin and smaller in the Brahmaputra basin. Further, socio-economic developments are expected to be strong in the Indus and Ganges, whereas in the Brahmaputra it is expected to be moderate. We will add a sentence to Section 2 Study Area to clarify why we selected the entire IGB as study domain.

*14. P5 L11: What about Tehri Dam, or various large dams in Arunachal?*
Tehri Dam and the various dams in Arunachal are located in the upstream domains.

*15. P8 L7: Meaning flood irrigation (as a technique), or surface water irrigation? Because surely large parts are irrigated from ground water (as a source).*
We mean flood irrigation. We have changed the text of the manuscript according to this comment.

*16. P8 L24: About 6 km*
A grid cell size of 5 arc-min corresponds with a grid cell size of ~ 9 km in the IGB.

*17. P9 L12: Water 'availability' also influenced by pollution. Return flows pollute rivers, and make the water unavailable?*
We agree with the reviewer that water availability can be influenced by pollution and that once it is polluted it makes the water unavailable for use. In our approach, we did not take the effects of water pollution on water availability into account. This can be considered as a limitation and might also introduce uncertainties in the outcomes. Therefore, we will add this point to the discussion.

*18. P10 L6: Please indicate the spatial resolution of IMAGE*
The spatial resolution of IMAGE is 5 x 5 arc-min. However we did not use this information since we only use land use change information at regional level (i.e. for India and the other parts of South Asia).

*19. P10 L9: What about the endogeneity of landuse? If water becomes scarce, landuse will change. Maybe 2nd order effect, but should be mentioned.*
Endogeneity of land use has been considered in our model approach. In the IMAGE model, the aim is to fulfil the food demand at regional level. If water scarcity limits the yield, land use change occurs by allocating more (rainfed) land to meet the food production needs. We assume however that both the crop production and crop types remain as they are, whereas, in reality, farmers can decide to switch to other crop types or crop varieties when crop growth conditions are not favourable any more (i.e. due to the higher risk for heat stress that is a consequence of increased temperature (extremes)). This will most likely influence the irrigation water requirements and subsequently the projected amount of water scarcity. We will add an extra sentence to clarify that we make assumptions on the crop distributions and the crop types and we will add this point to the discussion to highlight the uncertainties that can emerge from these assumptions.

*20. P11 L2: This is too lengthy a period, during which water resource use in this region has undergone huge changes. To quote a single average for this entire period (for any variable) is highly problematic.*
We refer to comment 4f.

*21. P11 L3: No way to confirm that this represents reality.*
Both the SPHY and LPJmL models have been validated in previous studies. The SPHY model has been calibrated and validated for the upstream domains in Wijngaard et al. (2017). The LPJmL model has been validated and tested for global applications, such as river discharge (Biemans et al., 2009), irrigation requirements (Rost et al., 2008), crop yields (Fader et al., 2010) and sowing dates (Waha et al., 2012). For South Asia, the model has been applied to study the adaptation potential of increased dam capacity and improved irrigation efficiency under changing climate conditions (Biemans et al.,

2013), and for the estimation of crop-specific seasonal irrigation water requirements (Biemans et al., 2016). In both studies the irrigation withdrawals have been validated for India and Pakistan. We will include this information in the Data and Methods section of the manuscript.

*22. P13 L2: Is this realistic at all?*
We stated erroneous values corresponding with relative changes in blue water consumption. The values listed (i.e. 24%/ 42%/ 107%) represent relative increases in blue water consumption that are projected for the mid of the 21$^{st}$ century (i.e. 2041-2070). The relative increases that are projected for the end of the 21$^{st}$ century (2071-2100) correspond with the following values: 36% for the Lower Indus Basin, 60% for the Lower Ganges Basin, and 147% for the Lower Brahmaputra Basin. We will update the manuscript by replacing the old values with new values.

   The values are difficult to compare with those from other studies, because regional studies investigating future changes in water consumption, using the RCP/SSP combinations, are lacking. To the authors' knowledge the only regional study implemented is the study of Gain and Wada (2014) in the Brahmaputra River Basin. Nevertheless, the authors of the cited study used the SRES A2 scenario framework to assess changes. The authors of the cited study project an approximate doubling of the blue water consumption in the Lower Brahmaputra between 2000 and 2050 under the SRES A2 scenario, which is often seen as similar to the RCP8.5-SSP3 scenario. These projected changes are in line with the projected changes for the Lower Brahmaputra in our study.

*23. P14 L15: This involves the value loaded assumption that all surface water can be abstracted and consumed, leaving nothing for instream flows and flows to the ocean. And to say that 'sustainable gw' is something in addition to surface flows is to forget that all GW recharge would normally (i.e. in the absence of abstraction) end up as discharge (=baseflow in rivers or in coastal aquifers as discharge directly to oceans). In other words, to count both is to double count.*
As mentioned under comment 4a, double counting does not occur, because groundwater withdrawals (i.e. abstraction from groundwater reservoirs, using (artificial) wells) prevent groundwater to be discharged as baseflow in rivers. This also implies that in the integrated modelling approach we use groundwater and surface water are not disconnected. When upstream groundwater withdrawals occur, baseflow reduces, which eventually affect downstream surface water availability. Further, in the LPJmL model it is possible to abstract all the available surface water locally, from neighbouring grid cells, the upstream located reservoirs or the canals before groundwater can be withdrawn. However, not all the water that is abstracted is also consumed. Water can be lost during conveyance by open water evaporation or as a return flow into the river network. After the application to the field, again only a part of the water will be used for evapotranspiration (i.e. blue water consumption). The remaining part will recharge groundwater or will discharge as a return flow to the river. We will update the model description to clarify that not all withdrawn water is consumed, and that a part of the withdrawn water is discharged in the river as a return flow.

*24. P16 L6: Will farming actually adapt to the shorter growing season or will it try to compensate for the higher temperature through extra irrigation?*
In LPJmL, the phenology of a crop (i.e. represented by a single period between sowing and crop maturity) is defined by the accumulated amount of growing degree days (GDD) a crop needs to reach physiological maturity. Thereby, the daily number of GDD is highly depending on temperature, and is defined as the difference between the daily average temperature and a crop-specific base temperature. This means under higher temperatures, the daily number of GDD is higher, and the accumulated amount of GDD (or heat units) necessary for crop maturity is reached earlier, but with lower yields. This means that the growing season for a specific crop becomes shorter. This will not necessarily influence the farmer decision to adapt. However higher temperatures also mean a higher risk for heat stress, which is therefore a likely reason for the farmer to adapt by compensating higher temperature through extra irrigation. We will add this point to the discussion.

*25. P16 L16: Unless the other studies used some 'maximum limit' on C as a fraction of A, the fact that they use C/A and this study uses C-A should make no difference to the results.*

We agree with the reviewer that the difference in water scarcity indicators among different studies should not make a difference in the water scarcity trends that are projected. However, in terms of absolute numbers the use of different water scarcity indicators among different studies can hamper the comparison of outcomes between those studies. Therefore, we emphasized that the differences between our study and other studies are related to the use of different modelling approaches and scenarios or the use of different water scarcity indicators.

*26. P17 L23: More generally, the cropping decisions in this model are not endogenously determined, which in fact they would be*
We refer to minor comment 19.

*27. P18 L15: Downstream DEPENDENCY woulld be large only if there is high use of the water that comes from the upstream. The fact that downstream AVAILABILITY is much higher than upstream availability is due to the terrain and rainfall pattern: I & G catchments in the Himalayas get high rain, whereas catchment of B which is in Tibet does not.*
Figure 3 of the manuscript shows the surface water availability for the four seasons that prevail in South Asia. It shows that especially during the melt season and the monsoon season the water availability is higher in the upstream regions of the Indus and Ganges basins than in the respective downstream regions (i.e. which can be less than 100 mm/year in the downstream regions of the Indus basin). This can mainly be attributed to the high melt contributions in the upstream Indus (more than 80% at Besham Qila; Wijngaard et al., 2017) and the high precipitation in the upstream Ganges. In the Brahmaputra basin, the difference in water availability between upstream and downstream domains is smaller, but still the upstream part receives over 3000 mm precipitation per year. It indicates that the downstream dependency on upstream water resources is large, especially in the Indus and Ganges basins. This is also indicated by the relative contributions of mountain water to the total discharge at the river outlets of the Indus, Ganges, and Brahmaputra. Relative contributions can reach up to about 80% at the outlets of the Indus and Brahmaputra, and up to about 60% at the outlet of the Ganges (Biemans et al., in prep.).

*28. P18 L16: Could this simply be an artefact of glaciers melting, thereby generating much more rapid runoff in the Himalayas?*
This cannot be seen as an artefact. The projections for the upstream domains show increases in ice melting and precipitation in the Indus river basins, and an increase in precipitation in the Ganges river basin. The increased ice melting and precipitation eventually result in increased runoff in the Himalayas.

*29. P18 L26: Isn't this within the uncertainty of the models?*
We refer to comment 3.

*30. P18 L29: Shouldn't this have been fairly obvious, the moment one considers the climate predictions for this region, and knowing the socio-economic trends in this region?*
Yes, it is should be obvious that socio-economic development is a key driver in the evolution of the South Asian water gap, whereas climate change works as a decelerator. Though it is obvious, there has not been a study so far, quantifying the impacts of climate change and/or socio-economic development on the evolution of the water gap in the IGB by using a high-resolution modelling approach that considers seasonal variations and which is forced by a set of (combined) climate change and socio-economic scenarios.

*31. P23 L5: Cannot find this reference. Is it published? Or submitted?*
The reference of Lutz et al. (under review) belongs to a publication that is currently under review.

*32. P28: Please indicate the source of this data.*
To derive this map, we have extracted data from the MIRCA2000 dataset (Biemans et al., 2016; Portmann et al., 2010). We will add a reference to the caption of the figure.

**References**

Biemans, H., Siderius, C., Lutz, A. F., Nepal, S., Ahmad, B., Hassan, T., von Bloh, W., Wijngaard, R. R., Wester, P., Shrestha, A. B. and Immerzeel, W. W.: Himalayan mountain water resources crucial for downstream agriculture, in prep.

Pellicciotti, F., Buergi, C., Immerzeel, W. W., Konz, M. and Shrestha, A. B.: Challenges and Uncertainties in Hydrological Modeling of Remote Hindu Kush–Karakoram–Himalayan (HKH) Basins: Suggestions for Calibration Strategies, Mt. Res. Dev., 32, 39–50, doi:10.1659/MRD-JOURNAL-D-11-00092.1, 2012.

---

## Author Comment (AC3) · 9 Jul 2018

We greatfully acknowledge the editor for his/her remarks and suggestions, which improved the quality of the manuscript significantly. We have carefully considered the suggestions of the editor and we provide a point-by-point response to the editor's comments. For clarity, the editor's comments are given in italics and the responses are given in plain text. The manuscript will be modified accordingly to the responses that are given to the comments.

*I agree with the major comments and concerns of the anonymous reviewer. In addition, I would like the authors to add a brief discussion in the "Uncertainties and Limitations" section on the possible non-stationarity in the hydrologic model parameters because of the change in climate and landuse.*

Thank you. We have added a brief discussion on the possible non-stationarity in the hydrological model parameters in the section "Uncertainties and Limitations".

---

## Referee Report (RR1)

Comments on:          "Climate change vs. socio-economic development: Understanding the future South-Asian water gap" (VERSION2: JULY 2018)

Authors:              Wijngaard et al.

Submitted to:         HESS

Article ref no.:      hess-2018-16

I have gone through the revised paper carefully.

1. The authors have clarified the doubts raised about the modelling. No problem there.

2. Regarding blue water gap (BWG) versus blue water scarcity (BWS): the authors have revised the manuscript to emphasize that they are really estimating the 'blue water gap' (with the assumption that there is no adaptive response to water shortages). But the text is now rather inconsistent—switching between the two terms. There is no real discussion (either in the para starting on p3 line23, or later in the methods section) of the implications of using BWG as a proxy for BWS—so no real recognition of the socially constructed nature of this outcome variable. The authors say that Figure 9 shows how BWG is especially significant in urban areas. But this actually illustrates the risk in using BWG as a proxy for scarcity: because in urban areas, whenever there is a 'gap', it will get addressed through more imports (because the quantities required are still relatively small and drinking water is a political priority). So in fact it is not much of an issue in urban areas!

3. The authors have taken pains to include the issue of Environmental Flows as an outcome variable and added a whole analysis there. This is laudable and has made the article potentially more interesting. Their definition of EFR for low flow season as being 60% of the mean monthly flow during that season is, however, quite stringent: this makes even the Brahmaputra as 'not meeting EFR needs even in current/reference scenario. This seems rather extreme and needs to be checked.

4. The biggest problem I continue have with the paper is "what does the paper contribute to our understanding?". The authors' response has been "we are doing more novel modelling with finer spatial scale, better modelling of many of the processes involved, etc. than anyone else has done". So here we are facing a fundamental difference in our understanding of what constitutes 'contribution' to scientific knowledge: I believe that a 'better' model is one which gives either more precise predictions of changes in some outcome variable(s) or counter-intuitive results on the 'sign' of change in the outcome variable. In other words, 'better' has to be judged not by modelling sophistication per se, but whether the sophistication generates results that are different (and robustly so) from what we already know.

When I look at their results carefully, the uncertainties are even higher and the non-results even sharper:

In 3 out of 4 scenarios, the blue-water gap actually DECREASES across all 3 basins and all seasons. But these decrease estimates have a high standard deviation, making most of them no different from zero.

In the 4th scenario (greatest climate change & highest economic/demographic growth rate) we see that socio-economic growth outstrips effects of CC and there is an INCREASE in the blue-water gap. HOWEVER, this increase is not statistically significant: it is 14% with a standard deviation of 43 (% points I presume). Which means it is not statistically different from 0 (the NO CHANGE scenario).

So after all this effort, they are not able to tell us even what the SIGN on the net change is going to be, let alone give us a robust estimate of the magnitude of net change.

So why is this whole (very technically competent) exercise useful?

---

## Author Response (AR2)

We gratefully acknowledge the reviewer for his/her remarks and suggestions, which improved the quality of the manuscript significantly. We have carefully considered the suggestions of the reviewer and we provide a point-by-point response to the reviewer's comments. For clarity, the reviewer's comments are given in italics and the responses are given in plain text. The manuscript will be modified accordingly to the responses that are given to the comments.

*I have gone through the revised paper carefully.*

Thank you. We have tried to address all concerns and our detailed response is provided below.

1. *The authors have clarified the doubts raised about the modelling. No problem there.*

Thank you.

2. *Regarding blue water gap (BWG) versus blue water scarcity (BWS): the authors have revised the manuscript to emphasize that they are really estimating the 'blue water gap' (with the assumption that there is no adaptive response to water shortages). But the text is now rather inconsistent— switching between the two terms. There is no real discussion (either in the para starting on p3 line23, or later in the methods section) of the implications of using BWG as a proxy for BWS—so no real recognition of the socially constructed nature of this outcome variable. The authors say that Figure 9 shows how BWG is especially significant in urban areas. But this actually illustrates the risk in using BWG as a proxy for scarcity: because in urban areas, whenever there is a 'gap', it will get addressed through more imports (because the quantities required are still relatively small and drinking water is a political priority). So in fact it is not much of an issue in urban areas!*

We agree that the use of both blue water gap and blue water scarcity is confusing, and in the revised manuscript we will only refer to blue water gap. The focus of this study is on assessing the combined impacts of climate change and socio-economic developments on the future blue water gap. The blue water gap as defined in our study is the amount of unsustainable groundwater that is withdrawn to fulfil the blue water demand. This means the blue water gap is only present when renewable or sustainable blue water resources (i.e. surface water and renewable groundwater) are not available (locally), which means that the blue water gap can be an indicator for a scarcity in renewable water resources. Within the LPJmL model the blue water gap is closed by (unsustainable) groundwater withdrawals. As a result of this model concept water scarcity, in stricto sensu, does not occur, because the demand is always met. We realize that we have not stated this clearly in our last response to the reviewer. Also, we realize that due to the use of the two terms "blue water gap" and "blue water scarcity" throughout the manuscript, the text has become inconsistent and unclear. To avoid the inconsistencies between the two terms we will rephrase the paragraphs in Section 1 Introduction, Section 4.6 Comparison with other studies, and Section 4.7 Uncertainties and Limitations and we will systematically refer to blue water gap only.

3. *The authors have taken pains to include the issue of Environmental Flows as an outcome variable and added a whole analysis there. This is laudable and has made the article potentially more interesting. Their definition of EFR for low flow season as being 60% of the mean monthly flow during that season is, however, quite stringent: this makes even the Brahmaputra as 'not meeting EFR needs even in current/reference scenario. This seems rather extreme and needs to be checked.*

Thank you, we also consider it as a valuable addition and we thank you for this suggestion. We use the well accepted Variable Month Flow (VMF) method of Pastor et al. (2014) to estimate environmental flow requirements (EFRs). This method enables to calculate monthly EFRs depending on the season (low, intermediate and high flow seasons). During the low flow season the EFR is 60% of the mean monthly flow (MMF), 45% of MMF during the intermediate flow season and 30% during the high flow season. Pastor et al. (2014) also compared five methods for assessing EFRs on global and local scales: the Smakhtin, Tennant, Tessmann, VMF, and $Q_{90}\_Q_{50}$ methods. Their review showed that the VMF and the Tessmann methods perform best. The difference between these two methods is that the Tessmann method allocates all water to EFRs in low flow seasons, whereas the VMF method allocates 60% of the MMF to EFRs in low flow seasons. Since the Tessmann method does not allow water withdrawals during the low flow season it is therefore less valid in a region like the IGB. Using the Tessmann method would result in an even more stringent EFR.

In the Brahmaputra River Basin, the low flow season coincides with the rabi season. During this season, the water consumption is significantly higher at most irrigated croplands than during the kharif season (Figure 5, 7, and 8). The combination of higher water consumptions and lower water availability explains why EFRs are not met during the low flow seasons and we therefore think this is realistic, and the VMF method is valid to use here.

4. *The biggest problem I continue have with the paper is "what does the paper contribute to our understanding?". The authors' response has been "we are doing more novel modelling with finer spatial scale, better modelling of many of the processes involved, etc. than anyone else has done". So here we are facing a fundamental difference in our understanding of what constitutes 'contribution' to scientific knowledge: I believe that a 'better' model is one which gives either more precise predictions of changes in some outcome variable(s) or counter-intuitive results on the 'sign' of change in the outcome variable. In other words, 'better' has to be judged not by modelling sophistication per se, but whether the sophistication generates results that are different (and robustly so) from what we already know.*

*When I look at their results carefully, the uncertainties are even higher and the non-results even sharper: In 3 out of 4 scenarios, the blue-water gap actually DECREASES across all 3 basins and all seasons. But these decrease estimates have a high standard deviation, making most of them no different from zero. In the 4th scenario (greatest climate change & highest economic/demographic growth rate) we see that socio-economic growth outstrips effects of CC and there is an INCREASE in the blue-water gap. HOWEVER, this increase is not statistically significant: it is 14% with a standard deviation of 43 (% points I presume). Which means it is not statistically different from 0 (the NO CHANGE scenario). So after all this effort, they are not able to tell us even what the SIGN on the net change is going to be, let alone give us a robust estimate of the magnitude of net change. So why is this whole (very technically competent) exercise useful?*

**Novelty**

In the last decades, several climate change impact assessments have been conducted in the mountainous regions and surrounding lowlands of river basins in South and Central Asia. For instance, Barnett et al. (2005) concluded by means of a global study that in the HKH region future water availability will most likely decrease during the dry season mainly due to vanishing glaciers and snow. Immerzeel et al. (2010) found that the importance of glaciers and snow is highly variable, and that projections in future river flow do not reveal a consistent pattern. A later study of Lutz et al. (2014) projected runoff increases in the upstream domains of the Ganges, Brahmaputra, Salween, Mekong, and Indus until 2050, which were primarily attributed to increases in precipitation and accelerated melt (i.e. only in the upstream domain

of the Indus). The assessments indicate that the understanding on the impacts of climate change on upstream water availability in the HKH region has improved over the last decades, despite the large uncertainties that are mainly related to climate change projections. Although scientific research in this field has clearly advanced, there are still large unknowns. For instance, to what extent the climate-induced changes in upstream hydrology will affect the downstream domains of the South Asian river basins, and how it relates to the anticipated socio-economic developments and related changes in water demand. The relevance and novelty of this study is therefore in understanding the link between climate change and socio-economic development with the aim to quantify how the future South Asian blue water gap will develop.

In addition, there are also clear disciplinary novel components of our study:

- For the first time a model is applied that integrates both fundamental cryospheric processes in the upstream part of the basins with an advanced water distribution model that includes reservoirs, multiple cropping cycles and irrigation that integrates extensive canal systems and groundwater. This model takes upstream-downstream links and lateral transport into consideration, which enables the possibility to assess the effects of changes in upstream water supply on downstream water availability and to improve analyses on the regional "blue" water gap.
- Further, the novelty is in the use of gridded socio-economic and land use scenarios, combined with an ensemble of downscaled and bias-corrected GCMs, to assess the combined impacts of climate change and socio-economic development on the future South Asian water gap.

We will emphasize these novelties more explicitly in the introduction.

**Uncertainties**

The outcomes of this study are prone to large uncertainties that are mainly originating from the uncertainties in climate change projections. This is the case for almost all hydrological impact studies and it is not something that is unique to our case. We consider it a strength of our study that we show the full range of possible futures including uncertainty in both climate change and socio-economic projections. GCMs have a limited skill in simulating the complex mountainous climate regimes of Central and South Asia (Lutz et al., 2016b; Seneviratne et al., 2012). We do the best currently possible by using an advanced envelope-based selection approach to select climate models also based on their skill to simulate the regional climate. For each RCP, we selected four GCMs that represent the four corner points of possible climate conditions (i.e. warm-wet, warm-dry, cold-wet, and cold-dry) and thus cover a wide range of possible future climate conditions. The variation in climate change projections between different GCMs is however large, which results in a large spread among the climate models. This spread propagates subsequently to the hydrological model outcomes. This is a general problem in many studies that are conducted in the region (e.g. Arnell and Lloyd-Hughes, 2014; Lutz et al., 2014; Moors et al., 2011; Wijngaard et al., 2017). The upcoming CMIP6 model archive (Eyring et al., 2016) might improve the outcomes of the studies by reducing the variation in climate change projections between the different GCMs. We will mention this in Section 4.7 Uncertainties and Limitations. Since we only used four climate models per RCP, we realize that the standard deviation is not a suitable proxy for the expression of the uncertainties in Table 2. It is not an uncertainty, but a range of possible futures. For this reason, we replaced the standard deviation values in Table 2 by the min-max range, also for consistency throughout the manuscript. Further, we use a colour scheme to indicate the number of climate-model related outcomes that show the same sign of change as the projected mean change, as an indicator for the trend's significance.

   The updated Table 2 shows that the range of possible futures is indeed large, but this is the spread in possible futures the climate models currently project and it is important to provide this message. For the RCP combinations, a clear decreasing trend is projected on annual basis and on seasonal basis as well. For RCP4.5-SSP1, the decreasing trend is still present, although the projected relative decreases are smaller than for the RCP combinations, which can be attributed to the projected increases in domestic and industrial water consumption. For RCP8.5-SSP3 the projected changes in water demand are so large

that climate change cannot compensate them anymore. Despite the large range among the model outcomes, the mean of the outcomes for RCP8.5-SSP3 still indicate that the blue water gap will increase. This means, in turn, that socio-economic development can be considered as a key driver in the future South Asian water gap.

**Table 2.** Projected changes in the annual and seasonal blue water gap of the Indus and Ganges river basins under present (1981-2010) and far-future (2071-2100; EOC) conditions for RCP4.5, RCP8.5, RCP4.5-SSP1, and RCP8.5-SSP3. The values between the parentheses represent the minimum and maximum projected changes in the blue water gap. The colours indicate the number of model runs (i.e. green: more than 3 runs; yellow: 2 runs; red: 1 run) that project the same sign of change as the projected mean change.

| Basin | Scenario | Annual | Winter | Pre-monsoon | Monsoon | Post-monsoon |
|---|---|---|---|---|---|---|
| Indus | REF [km$^3$] | 83 | 19 | 27 | 28 | 10 |
| | RCP45 EOC [%] | -36 (-59/-19) | -47 (-70/-32) | -35 (-61/-18) | -32 (-53/-13) | -33 (-54/-13) |
| | RCP85 EOC [%] | -37 (-52/-15) | -52 (-59/-35) | -44 (-49/-35) | -23 (-58/14) | -29 (-44/-10) |
| | RCP45-SSP1 EOC [%] | -21 (-50/1) | -31 (-60/-11) | -21 (-53/0) | -16 (-42/7) | -15 (-42/9) |
| | RCP85-SSP3 EOC [%] | 7 (-18/42) | -11 (-25/19) | -9 (-16/5) | 30 (-27/91) | 18 (-8/48) |
| Ganges | REF [km$^3$] | 35 | 13 | 10 | 6 | 6 |
| | RCP45 EOC [%] | -52 (-85/-26) | -61 (-89/-41) | -51 (-86/-23) | -44 (-82/-19) | -41 (-81/-8) |
| | RCP85 EOC [%] | -55 (-73/-23) | -66 (-78/-38) | -63 (-73/-45) | -39 (-72/14) | -34 (-64/10) |
| | RCP45-SSP1 EOC [%] | -23 (-74/16) | -37 (-79/-4) | -23 (-74/19) | -9 (-66/28) | -8 (-67/41) |
| | RCP85-SSP3 EOC [%] | 14 (-26/82) | -11 (-40/49) | 1 (-23/44) | 55 (-17/165) | 50 (-11/131) |